# VERINA: BENCHMARKING VERIFIABLE CODE GENERATION

**Zhe Ye**[1], **Zhengxu Yan**[1], **Jingxuan He**[1], **Timothe Kasriel**[1], **Kaiyu Yang**[2*], **Dawn Song**[1]
[1]University of California, Berkeley, [2]Meta FAIR

## ABSTRACT

Large language models (LLMs) are increasingly integrated in software development, but ensuring correctness in LLM-generated code remains challenging and often requires costly manual review. *Verifiable code generation*—jointly generating code, specifications, and proofs of code-specification alignment—offers a promising path to address this limitation and further unleash LLMs' benefits in coding. Yet, there exists a significant gap in evaluation: current benchmarks often focus on only individual components rather than providing a holistic evaluation framework of all tasks. In this paper, we introduce VERINA (Verifiable Code Generation Arena), a high-quality benchmark enabling a comprehensive and modular evaluation of code, specification, and proof generation as well as their compositions. VERINA consists of 189 manually curated coding tasks in Lean, with detailed problem descriptions, reference implementations, formal specifications, and extensive test suites. Our extensive evaluation of state-of-the-art LLMs reveals significant challenges in verifiable code generation, especially in proof generation, underscoring the need for improving LLM-based theorem provers in verification domains. The best model, OpenAI o3, achieves a 72.6% code correctness rate, 52.3% for specification soundness and completeness, and a mere 4.9% proof success rate (based on one trial per task). We hope VERINA will catalyze progress in verifiable code generation by providing a rigorous and comprehensive benchmark. We release our dataset on https://huggingface.co/datasets/sunblaze-ucb/verina and our evaluation code on https://github.com/sunblaze-ucb/verina.

## 1 INTRODUCTION

Large language models (LLMs) have shown strong performance in programming (Jain et al., 2025; Jimenez et al., 2024; Chen et al., 2021) and are widely adopted in tools like Cursor and GitHub Copilot to boost developer productivity (Kalliamvakou). LLM-generated code is becoming prevalent in commercial software (Peters, 2024) and may eventually form a substantial portion of the world's code. However, due to their probabilistic nature, LLMs alone cannot provide formal guarantees for the generated code. As a result, the generated code often contains bugs, such as functional errors (Wang et al., 2025) and security vulnerabilities (Pearce et al., 2022). When LLM-based code generation is increasingly adopted, these issues can become a productivity bottleneck, as they typically require human review to be resolved (Finley). Formal verification presents a promising path to establish correctness guarantees in LLM-generated code but has traditionally been limited to safety-critical applications due to high cost (Gu et al., 2016; Leroy et al., 2016; Bhargavan et al., 2013). Similarly to how they scale up code generation, LLMs have the potential to significantly lower the barrier of formal verification. By jointly generating code, formal specifications, and formal proofs of alignment between code and specifications, LLMs can offer higher levels of correctness assurance and automation in software development. This approach represents an emerging programming paradigm known as *verifiable code generation* (Sun et al., 2024; Yang et al., 2024).

Given the transformative potential of verifiable code generation, it is crucial to develop suitable benchmarks to track progress and guide future development. This is challenging because verifiable code generation involves three interconnected tasks: code, specification, and proof generation. We need to curate high-quality samples and establish robust evaluation metrics for each individual

---

*All data processing and experiments were conducted outside Meta.

Table 1: A comparison of VERINA with related prior works on LLMs for code generation and verification. We characterize whether each work supports the three foundational tasks for end-to-end verifiable code generation: CodeGen, SpecGen, ProofGen (Section 4.1). ● means fully supported, ◐ means partially supported, ○ means unsupported. If ProofGen is supported, we specify the proving style: automated theorem proving (ATP) or interactive theorem proving (ITP). For works supporting multiple tasks, we annotate if these tasks are supported in a modular and composable manner. Overall, VERINA offers more comprehensive and high-quality benchmarking compared to prior works.

| | | CodeGen | SpecGen | ProofGen | Proving Style | Compositionality | Language |
|---|---|---|---|---|---|---|---|
| Benchmarks | HumanEval (Chen et al., 2021), MBPP (Austin et al., 2021) | ● | ○ | ○ | – | – | Python |
| | Dafny-Synthesis (Misu et al., 2024) | ● | ◐ | ● | ATP | ✗ | Dafny |
| | DafnyBench (Loughridge et al., 2025) | ○ | ○ | ● | ATP | – | Dafny |
| | miniCodeProps (Lohn & Welleck, 2024) | ○ | ○ | ● | ITP | – | Lean |
| | FVAPPS (Dougherty & Mehta, 2025) | ● | ○ | ● | ITP | ✗ | Lean |
| Techniques | nl2postcond (Endres et al., 2024) | ○ | ● | ○ | – | – | Python, Java |
| | Clover (Sun et al., 2024) | ● | ● | ● | ATP | ✗ | Dafny |
| | AlphaVerus (Aggarwal et al., 2024) | ● | ○ | ● | ATP | ✗ | Rust |
| | AutoSpec (Wen et al., 2024) | ○ | ● | ● | ATP | ✗ | C/C++ |
| | SpecGen (Ma et al., 2025) | ○ | ● | ● | ATP | ✗ | Java |
| | SAFE (Chen et al., 2025) | ○ | ○ | ● | ATP | ✗ | Rust |
| | AutoVerus (Yang et al., 2025) | ○ | ○ | ● | ATP | – | Rust |
| | Laurel (Mugnier et al., 2025) | ○ | ○ | ● | ATP | – | Dafny |
| | Pei et al. (2023) | ○ | ○ | ● | ATP | – | Java |
| | Baldur (First et al., 2023), Selene (Zhang et al., 2024) | ○ | ○ | ● | ITP | – | Isabelle |
| | Rango (Thompson et al., 2025), PALM (Lu et al., 2024) | ○ | ○ | ● | ITP | – | Coq |
| | VERINA | ● | ● | ● | ITP | ✓ | Lean |

task, while also composing individual tasks to reflect real-world end-to-end usage scenarios where LLMs automate the creation of verified software directly from high-level requirements. Existing benchmarks, as discussed in Section 2, fall short as they lack comprehensive support for all three tasks (Loughridge et al., 2025; Aggarwal et al., 2024; Chen et al., 2025), quality control (Dougherty & Mehta, 2025), robust metrics (Misu et al., 2024), or a modular design (Sun et al., 2024).

To bridge this gap, we introduce VERINA (Verifiable Code Generation Arena), a high-quality benchmark to comprehensively evaluate verifiable code generation. It consists of 189 programming challenges with detailed problem descriptions, code, specifications, proofs, and comprehensive test suites. We format these problems in Lean (Moura & Ullrich, 2021), a general-purpose programming language with a rapidly growing ecosystem and applications in both formal mathematics (Mathlib community, 2020; Mathlib Community, 2022) and verification (de Medeiros et al., 2025a; Hietala & Torlak, 2024). Lean has become the one of the most popular platforms for LLM-assisted theorem-proving and verification, demonstrated by breakthrough results like AlphaProof (Google DeepMind, 2024) and production adoption at organizations like AWS (de Moura), with ongoing efforts to use Lean for verifying mainstream languages like Rust (Ho & Protzenko, 2022). We provide additional discussion on the choice of Lean in Appendix A.

VERINA is constructed with careful quality control. It draws problems from various sources, including MBPP (Misu et al., 2024; Austin et al., 2021), LiveCodeBench (Jain et al., 2025), and LeetCode, offering a diverse range of difficulty levels. All samples in the benchmark are manually inspected and revised to ensure clear text descriptions and accurate formal specifications and code implementations. Moreover, each sample also includes a comprehensive test suite with both positive and negative cases, which achieves 100% code coverage and passes the ground truth specification.

VERINA facilitates the evaluation of code, specification, and proof generation, along with flexible combinations of these individual tasks. We utilize the standard pass@$k$ metric (Fan et al., 2024) with our comprehensive test suites to evaluate code generation. For proof generation, we use the Lean compiler to automatically verify their correctness. Furthermore, we develop a multi-stage evaluation pipeline that systematically assesses model-generated specifications by combining theorem proving and comprehensive testing, providing a practical and robust way to score their soundness and completeness against our ground truth specifications.

The high-quality samples and robust metrics of VERINA establish it as a rigorous platform for evaluating verifiable code generation. On VERINA, we conduct a thorough experimental evaluation of ten state-of-the-art general-purpose LLMs and three LLMs or agentic frameworks specialized in theorem proving. Our results reveal that even the top-performing general-purpose LLM, OpenAI o3 (OpenAI), struggles with verifiable code generation, producing only 72.6% correct code solutions, 52.3% sound and complete specifications, and 4.9% successful proof in one trial. Among theorem-proving LLMs, the best model, Goedel Prover V2 32B (Lin et al., 2025), achieved an 11.2% proof

```
1   -- Description of the coding problem in natural language
2   -- Remove an element from a given array of integers at a specified index. The resulting array should
3   -- contain all the original elements except for the one at the given index. Elements before the
4   -- removed element remain unchanged, and elements after it are shifted one position to the left.
5   -- Code implementation
6   def removeElement (s : Array Int) (k : Nat) (h_precond : removeElement_pre s k) : Array Int :=
7     s.eraseIdx! k
8   -- Pre-condition
9   def removeElement_pre (s : Array Int) (k : Nat) : Prop :=
10    k < s.size -- the index must be smaller than the array size
11  -- Post-condition
12  def removeElement_post (s : Array Int) (k : Nat) (res: Array Int) (h_precond : removeElement_pre s k)
13      : Prop :=
14    res.size = s.size - 1 ∧ -- Only one element is removed
15    (∀ i, i < k → res[i]! = s[i]!) ∧ -- The elements before index k remain unchanged
16    -- The elements after index k are shifted by one position
17    (∀ i, i < res.size → i ≥ k → res[i]! = s[i + 1]!)
18  -- Proof (proof body omitted for brevity)
19  theorem removeElement_spec (s: Array Int) (k: Nat) (h_precond : removeElement_pre s k) :
20    removeElement_post s k (removeElement s k h_precond) h_precond := by sorry
21  -- Test cases
22  (s : #[1, 2, 3, 4, 5]) (k : 2) (res : #[1, 2, 4, 5]) -- Positive test with valid inputs and output
23  -- Negative test cases
24  (s : #[1, 2, 3, 4, 5]) (k : 5) -- Inputs violate the pre-condition at Line 12
25  (s : #[1, 2, 3, 4, 5]) (k : 2) (res : #[1, 2, 4]) -- Output violates the post-condition at Line 16
26  (s : #[1, 2, 3, 4, 5]) (k : 2) (res : #[2, 2, 4, 5]) -- Output violates the post-condition at Line 17
27  (s : #[1, 2, 3, 4, 5]) (k : 2) (res : #[1, 2, 4, 4]) -- Output violates the post-condition at Line 18
```

Figure 1: An example instance of VERINA, consisting of a problem description, code implementation, specifications (pre-condition and post-condition), a proof (optional), and comprehensive test cases. Note that we select this instance for presentation purposes and VERINA contains more difficult ones.

success rate in one trial. Interestingly, iterative refinement using Lean compiler feedback can increase the proof success rate to 20.1% with 64 refinement steps. However, this approach significantly raises costs and the success rate remains low. These findings underscore the challenges of verifiable code generation and highlight the critical role of VERINA in advancing the field.

## 2  BACKGROUND AND RELATED WORK

We present works closely related to ours in Table 1 and discuss them in detail below.

**Task support for verifiable code generation.** Writing code, specifications, and proofs for a verified software component is time-consuming when done manually. Although various studies have explored using LLMs to automate these tasks, they primarily focus on individual aspects, failing to capture the full spectrum of verifiable code generation. Benchmarks like HumanEval (Chen et al., 2021) and MBPP (Austin et al., 2021) have sparked impressive progress on LLM-based code generation but do not handle formal specifications or proofs. Many verification-focused efforts target only one or two tasks, while assuming the other elements are provided by the human user. For example, DafnyBench (Loughridge et al., 2025) and miniCodeProps (Lohn & Welleck, 2024) are two benchmarks designed exclusively for proof generation. Moreover, AutoSpec (Wen et al., 2024) and SpecGen (Ma et al., 2025) infer specifications and proofs from human-written code.

To the best of our knowledge, Dafny-Synthesis (Misu et al., 2024) and Clover (Sun et al., 2024) are the only two works that cover all three tasks, like VERINA. However, they target automated theorem proving using Dafny (Leino, 2010), while VERINA leverages interactive theorem proving in Lean. Moreover, they have relatively small numbers of human-written samples (50 and 62 respectively). In contrast, VERINA provides 189 high-quality samples that are manually validated and undergo rigorous quality assurance (Section 3.2).

**Automated and interactive theorem proving.** A major challenge in formal verification and verifiable code generation lies in tooling. Verification-oriented languages like Dafny (Leino, 2010) and Verus (Lattuada et al., 2023) leverage SMT solvers for automated theorem proving (De Moura & Bjørner, 2008; Barrett & Tinelli, 2018) and consume only proof hints, such as loop invariants (Pei et al., 2023) and assertions (Mugnier et al., 2025). However, SMT solvers handle only limited proof domains and behave as black boxes, which can make proofs brittle and hard to debug (Zhou et al., 2023). Interactive theorem proving (ITP) systems like Lean provide a promising target for verifiable code generation with LLMs. ITPs support constructing proofs with explicit intermediate steps. This visibility enables LLMs to diagnose errors, learn from unsuccessful steps, and iteratively refine their proofs. Recent work shows that LLMs can generate proofs at human level in math competi-

tions (Google DeepMind, 2024). Prior verification benchmarks in Lean include miniCodeProps (Lohn & Welleck, 2024) and FVAPPS (Dougherty & Mehta, 2025). miniCodeProps translates 201 Haskell programs and their specifications into Lean but is designed for proof generation only. FVAPPS contains 4,715 Lean programs with LLM-generated specifications from a fully automated pipeline that lacks human validation and quality control. In contrast, VERINA provides human-verified samples and captures all three foundational tasks in verifiable code generation.

**Task compositionality.** A key strength of VERINA is its modular design, which enables flexible evaluation of not only individual tasks but also their combinations (Section 4.2). This compositionality captures diverse real-world scenarios—from specification-guided code generation to end-to-end verifiable code generation—enabling a comprehensive assessment of different aspects of verifiable code generation. This modularity also facilitates targeted research on specific weaknesses, such as improving proof generation. On the contrary, all other prior works lack full compositionality. For example, Dafny-Synthesis (Misu et al., 2024) and Clover (Sun et al., 2024) mix specification and proof generation into a single task, lacking support for separate evaluation of each.

## 3 VERINA: DATA FORMAT, CONSTRUCTION, AND QUALITY ASSURANCE

We describe the VERINA benchmark, its data construction pipeline, and quality assurance measures.

### 3.1 OVERVIEW AND DATA FORMAT

VERINA consists of 189 standalone programs, annotated with natural language descriptions, code, specifications, proofs, and test cases. The code, specification, and proof are all written in Lean. An example is illustrated in Figure 1, consisting of:

- *Natural language description (Line 1–4)*: informal description of the programming problem, capturing the intent of the human developer.

- *Code (Line 5–7)*: ground truth code implementation that solves the programming problem.

- *Specification (Line 8–17)*: ground truth formal specification for the programming problem. It consists of a pre-condition, which states properties the inputs must satisfy, and a post-condition, which states desired relationship between inputs and outputs.

- *Proof (Optional, Line 18–20)*: formal proof establishing that the code satisfies the specification. Ground truth proofs are optional in VERINA, as they are not required for evaluation. Model-generated proofs can be checked by Lean directly. Nevertheless, we invest significant manual effort in writing proofs for 46 out of 189 examples as they help quality assurance (Section 3.2).

- *Test suite (Line 21–27)*: a comprehensive suite of both positive and negative test cases. Positive tests are valid input-output pairs that meet both the pre-condition and the post-condition. Negative tests are invalid inputs-output pairs, which means either the inputs violate the pre-condition or the output violates the post-condition. These test cases are useful for evaluating model-generated code and specifications, as detailed in Section 4.1. They are formatted in Lean during evaluation.

**Benchmark statistics.** Table 2 presents key statistics of VERINA. Natural language descriptions have a median length of 110 words, ensuring they are both informative and detailed. Code ranges up to 38 lines and specifications up to 62 lines, demonstrating that VERINA captures complex tasks. With a median of 5 positive tests and 12 negative tests per instance, the constructed test suites provide strong evidence for the high quality and correctness of VERINA.

Table 2: Statistics of VERINA.

| Metric | Median | Max |
|---|---|---|
| # Words in Description | 110 | 296 |
| LoC for Code | 9 | 38 |
| LoC for Spec. | 4 | 62 |
| # Positive Tests | 5 | 13 |
| # Negative Tests | 12 | 27 |

### 3.2 BENCHMARK CONSTRUCTION AND QUALITY ASSURANCE

VERINA consists of 189 problems sourced from different origins. We employ a meticulous data curation process that combines careful translation, thorough manual review, and automated mechanisms, leading to a rigorous and high-quality benchmark for verifiable code generation.

To construct VERINA, we first consider MBPP-DFY-50 (Misu et al., 2024) as our data source. It consists of MBPP (Austin et al., 2021) coding problems paired with human-verified solutions in Dafny. Each instance contains a natural language problem description, code implementation, specifications, proof, and test cases. We manually translated 49 problems into Lean, refining and verifying each translation. To extend the benchmark, we added 59 more human-authored Dafny instances from CloverBench (Sun et al., 2024). These were translated into Lean using OpenAI o3-mini with few-shot prompting based on our manual translations, followed by manual inspection and correction.

Additionally, VERINA incorporates problems adapted from student submissions to a lab assignment in a course on theorem proving and program verification. Students, both undergraduate and graduate, were encouraged to source problems from platforms like LeetCode or more challenging datasets such as LiveCodeBench (Jain et al., 2025). They formalized and solved these problems in Lean, providing all necessary elements in VERINA's format (Section 3.1). We carefully selected the most suitable and high-quality submissions, resulting in 81 benchmark instances. In addition, we manually reviewed and edited the submissions to ensure their correctness. During our evaluation, we observe problems adapted from student submissions are generally more difficult than problems translated from Dafny datasets on all models, with detailed analysis provided in Appendix D.

**Quality assurance.** During the data collection process, we consistently enforce various manual and automatic mechanisms to ensure the high quality of VERINA:

- *Detailed problem descriptions*: The original problem descriptions, such as those from MBPP-DFY-50, can be short and ambiguous, making them inadequate for specification generation. To resolve this, we manually enhanced the descriptions by clearly outlining the high-level intent, specifying input parameters with explicit type information, and detailing output specifications.

- *Full code coverage with positive tests*: Beyond the original test cases, we expanded the set of positive tests to ensure that they achieve full line coverage on the ground truth code. We created these additional tests both manually and with LLMs. We leveraged the standard `coverage.py` tool to verify complete line coverage, since Lean lacks a robust coverage tool. This approach aligns with common practices for assessing functional correctness across languages (Cassano et al., 2023; Roziere et al., 2022). For Python reference implementations, we either used the original MBPP code or generated an implementation from the enhanced problem description via OpenAI's o4-mini with manual validation. To further ensure coverage transferability, we manually inspected all benchmark instances and confirmed that our test suites also achieve 100% line coverage on the Lean ground truth implementations.

- *Full test pass rate on ground truth implementations and specifications*: We evaluated both the ground truth implementations and specifications against our comprehensive test suites. All ground truth implementations and specifications successfully pass their respective positive tests, confirming the quality of the implementations and specifications in VERINA.

- *Necessary negative tests*: We mutated each positive test case to construct at least three different negative tests that violate either the pre- or the post-condition, except when the function's output has boolean type, in which case only a single negative test can be created. These negative tests are explicitly categorized based on whether they violate the pre-condition or the post-condition to enable separate and precise evaluation of each specification component. We made sure that our ground truth code and specifications do not pass these negative tests.

- *Preventing trivial code generation*: VERINA allows providing ground truth specifications as an optional input for the code generation task (discussed in Section 4.1). We crafted all ground truth specifications such that they cannot be directly used to solve the coding problem. This prevents LLMs from generating an implementation trivially equivalent to the specification. As a result, the model must genuinely demonstrate semantic comprehension of the reference specification and non-trivial reasoning to generate the corresponding implementation.

- *Manual review and edits*: Each benchmark instance was manually reviewed by at least two authors, carefully inspecting and editing them to ensure correctness and high quality.

## 4    EVALUATING VERIFIABLE CODE GENERATION USING VERINA

VERINA enables comprehensive evaluation of verifiable code generation, covering foundational tasks—code, specification, and proof generation—and their combinations to form an end-to-end

pipeline from natural language descriptions to verifiable code. We also introduce a novel framework for a reliable automatic evaluation of model-generated specifications.

## 4.1 FOUNDATIONAL TASKS AND METRICS

As shown in Figure 2, all three foundational tasks include natural language descriptions and function signatures (Lines 7, 11, and 15 in Figure 1) as model inputs, which captures human intent and enforces consistent output formats, facilitating streamlined evaluation.

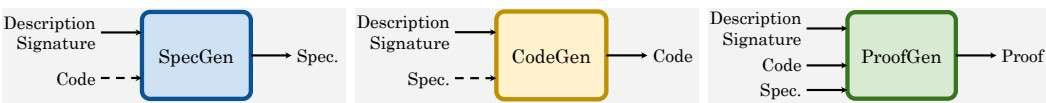

Figure 2: VERINA's three foundational tasks. Dashed arrows represent optional inputs.

**Specification generation (SpecGen).** Given a description, signature, and *optionally* code implementation, the model generates a formal specification. Next, we formally define the soundness and completeness relationships between the generated specification and the ground truth specification. Then, we describe our multi-stage evaluation pipeline to assess whether these relationships hold.

Let $\phi$ denote the set of programs that satisfy the ground truth specification and $\hat{\phi}$ the set that align with the generated specification. An ideal generated specification should achieve $\hat{\phi} = \phi$, which entails two properties—(i) *soundness* ($\hat{\phi} \subseteq \phi$): it is "small enough" to cover only correct programs, and (ii) *completeness* ($\phi \subseteq \hat{\phi}$): it is "large enough" to cover all correct programs. Since specifications consist of pre-conditions and post-conditions, let $P$ and $\hat{P}$ denote the ground truth and model-generated pre-conditions, respectively, and $Q$ and $\hat{Q}$ the corresponding post-conditions. In VERINA, we define the soundness and completeness of $\hat{P}$ and $\hat{Q}$ as follows:

- $\hat{P}$ is sound iff $\forall \overline{x}. P(\overline{x}) \Rightarrow \hat{P}(\overline{x})$, where $\overline{x}$ are the program's input values. Given the same post-condition (e.g., $Q$), it is more difficult for a program to satisfy $\hat{P}$ than $P$. This is because $\hat{P}$ allows more inputs, which the program must handle to meet the post-condition. As a result, the set of programs accepted by $\hat{P}$ a subset of those accepted by $P$.

- $\hat{P}$ is complete iff $\forall \overline{x}. \hat{P}(\overline{x}) \Rightarrow P(\overline{x})$. Given the same post-condition, the set of programs accepted by $\hat{P}$ is now a superset of those accepted by $P$, since $\hat{P}$ is more restrictive than $P$.

- $\hat{Q}$ is sound iff $\forall \overline{x}, y. P(\overline{x}) \wedge \hat{Q}(\overline{x}, y) \Rightarrow Q(\overline{x}, y)$, where $y$ is the output value. For any valid inputs w.r.t. $P$, the set of output accepted by $\hat{Q}$ is a subset of those accepted by $Q$, establishing soundness.

- Symmetrically, $\hat{Q}$ is complete iff $\forall \overline{x}, y. P(\overline{x}) \wedge Q(\overline{x}, y) \Rightarrow \hat{Q}(\overline{x}, y)$.

To practically and reliably assess whether the above relationships hold, we develop a multi-stage evaluator based on theorem proving and comprehensive testing, as shown in Figure 3. Given a soundness or completeness relationship $R$, the evaluator first attempts to prove $R$ using LLM-based theorem provers, as they provide formal guarantees when proof is successful. When the prover is inconclusive, e.g. due to the incapability of current LLM-based provers (as detailed in Appendix C.5), the evaluator proceeds with a practical testing-based framework using our comprehensive test suites. In this testing-based process, we check $R$ against concrete values in test cases. Specifically, we distinguish between negative tests that violate pre-conditions and those that violate post-conditions, applying them separately to evaluate the corresponding specification component.

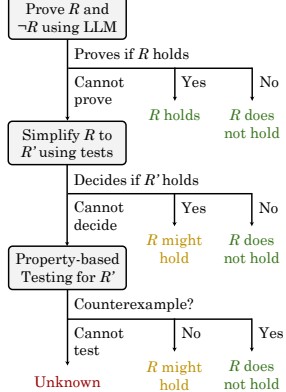

Figure 3: Our evaluator for specification generation.

For example, to evaluate $\hat{Q}$'s soundness, we check if $P(\overline{x}) \wedge \hat{Q}(\overline{x}, y) \Rightarrow Q(\overline{x}, y)$ holds for all test cases $(\overline{x}, y)$ in our test suite. We denote this simplified version of $R$ as $R'$. For many cases, e.g., the specification in Figure 1, Lean can automatically determine if $R'$ holds (Selsam et al., 2020) and we return the

corresponding result. Otherwise, we employ property-based testing with the `plausible` tactic in Lean (Lean Prover Community, 2024). It generates diverse inputs specifically targeting the remaining universally and existentially quantified variables in $R'$, systematically exploring the space of possible values to test $R'$. In Appendix C.5, we provide a detailed description of how we implement these metrics in Lean.

Since our evaluator integrates proof and testing, it can certify $R$ holds when a formal proof of $R$ succeeds, and it can certify $R$ does not hold by producing counterexamples. When only testing passes without a proof, the evaluator returns $R$ might hold, reflecting strong empirical evidence that $R$ holds. While it cannot formally establish $R$ holds, it remains highly robust in this regard, due to our comprehensive test suite with both positive and negative tests, which achieve full coverage on ground truth code implementations. Lean's property-based testing cannot handle a small number of complicated relationships on some testcases, for which our evaluator returns unknown. To further enhance the accuracy of our metric, we repeat our evaluation framework in Figure 3 to check $\neg R$. We compare the evaluator outcomes on $R$ and $\neg R$, selecting the definitive result whenever the other yields unknown.

Our final metrics for SpecGen include individual pass@$k$ scores (Chen et al., 2021) for soundness and completeness of all generated pre-conditions and post-conditions, as well as aggregated scores that soundness and completeness hold simultaneously for pre-condition, post-condition, and the complete specification. Since our specification evaluator may return unknown, we plot error bars indicating the lower bound (treating unknown as $R$ does not hold) and upper bound (treating as $R$ holds).

To illustrate our metric, consider the ground truth pre-condition `k < s.size` at Line 12 of Figure 1, and model-generated pre-condition `k < s.size - 1` and `k < s.size + 1`. `k < s.size - 1` can be determined as unsound using the positive test `(s : #[1, 2, 3, 4, 5]) (k : 4)`, while `k < s.size + 1` is incomplete based on the negative test `(s : #[1, 2, 3, 4, 5]) (k : 5)`. We provide more examples of our metrics for specification generation in Appendix E.

**Code generation (CodeGen).** Given a natural language description, function signature, and *optionally* specification, the model generates code implementing the desired functionality. Following standard practice, we evaluate the generated code by running it against positive test cases in VERINA and reporting the pass@$k$ metric defined by Chen et al. (2021). In Section 4.2, we will explore evaluating the code by proving its correctness with respect to the formal specification.

**Proof generation (ProofGen).** Given a description, signature, code, and specification, the model generates a formal proof in Lean to establish that the code satisfies the specification. This task evaluates the model's ability to reason about code behavior and construct logically valid arguments for correctness. We use Lean to automatically check the validity of generated proofs, and proofs containing placeholders (e.g., the `sorry` tactic) are marked as incorrect.

## 4.2 TASK COMBINATIONS

VERINA enables combining the three foundational tasks to evaluate various capabilities in verifiable code generation. These combined tasks reflect real-world scenarios where developers utilize the model to automatically create verified software in an end-to-end manner. Such modularity and

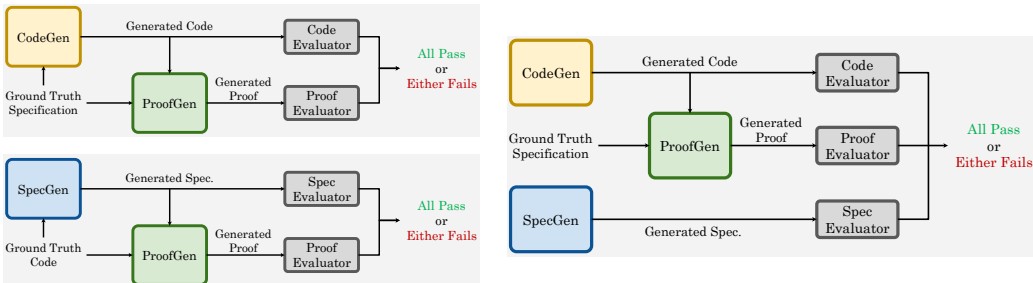

Figure 4: Combinations of VERINA's foundational tasks: specification-guided code generation (*top left*), specification inference from code (*bottom left*), and end-to-end verifiable code generation (*right*). Natural language descriptions and function signatures are omitted in the figure for brevity.

compositionality highlight the generality of VERINA, which encompasses various tasks studied in previous work (Table 1). Three examples of combined tasks are (Figure 4):

- *Specification-Guided Code Generation*: Given a natural language description, function signature, and the *ground truth* specification, the model first generates the code and then proves that the code satisfies the specification. This aligns with tasks explored in FVAPPS (Dougherty & Mehta, 2025) and AlphaVerus (Aggarwal et al., 2024).

- *Specification Inference from Code*: Developers may have the code implementation and want the model to annotate it with a formal specification and prove their alignment. This corresponds to the setting in AutoSpec (Wen et al., 2024), SpecGen (Ma et al., 2025), and SAFE (Chen et al., 2025).

- *End-to-End Verifiable Code Generation*: For an even higher degree of automation, developers might start with only a natural language problem description and instruct the model to generate code and specification independently, and then generate the proof. This captures the scenario in Dafny-Synthesis (Misu et al., 2024) and Clover (Sun et al., 2024). In this task, we specifically require the model to generate a proof that the generated code satisfies the ground truth specification. This prevents the model from generating definitionally equivalent code and specifications to trivialize the proof, ensuring the evaluation reflects the model's true verification capability.

In these task combinations, a crucial design consideration is the dependency between code and specification. For example, in specification-guided code generation, it is important to assess how beneficial the ground truth specification is beyond the natural language description, which already captures the developer's intent. Additionally, for end-to-end verifiable code generation, it is essential to decide the order of the CodeGen and SpecGen modules—whether to make SpecGen dependent on the output of CodeGen, place SpecGen before CodeGen, or run them independently (as in Figure 4). We experimentally explore these design choices using VERINA in Section 5. Concurrent with our work, CLEVER (Thakur et al., 2025) introduces 161 manually crafted problems sourced from HumanEval (Chen et al., 2021) with ground truth specifications. However, CLEVER only supports the SpecGen task and the specification-guided code generation setting and cannot capture the full spectrum of workflows that VERINA enables through both individual and compositional tasks. We provide detailed comparison in Appendix C.4.

## 5 EXPERIMENTAL EVALUATION

**Experimental setup.** We evaluate a diverse set of ten state-of-the-art general-purpose LLMs and three LLMs or agentic frameworks specialized in theorem proving. We leverage 2-shot prompting to enhance output format adherence, with the 2-shot examples excluded from the final benchmark. For each task, we primarily report the pass@1 metric (Chen et al., 2021). We provide detailed input prompts, output formats, and LLM setups in Appendix C.

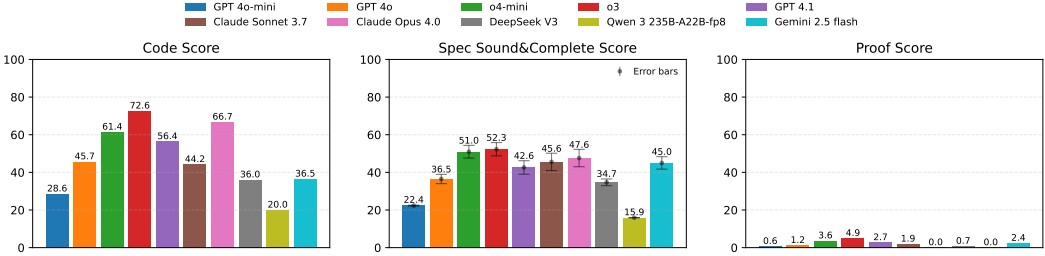

Figure 5: pass@1 performance of LLMs on VERINA's three foundational tasks.

**All foundational tasks are challenging, especially ProofGen.** As shown in Figure 5, code generation generally achieves the highest success rates across models, followed by specification generation, while proof generation remains the most challenging with pass@1 rates below 4.9% for all general purpose models. All three tasks pose significant challenges for current general purpose LLMs, with constructing Lean proofs that the implementation satisfies the specification being particularly hard and requiring specialized theorem proving capabilities. This also means that for any combined task involving ProofGen, e.g., the ones in Section 4.2, LLMs' performance will be heavily bottlenecked by the ProofGen subtask. Among the evaluated models, o4-mini, o3, Claude Sonnet 3.7, Claude

Opus 4.0, and Gemini 2.5 Flash demonstrate relatively stronger performance across tasks. We report detailed results on pre-condition and post-condition soundness and completeness in Appendix D, where we observe that generating sound and complete post-conditions is generally more difficult than pre-conditions.

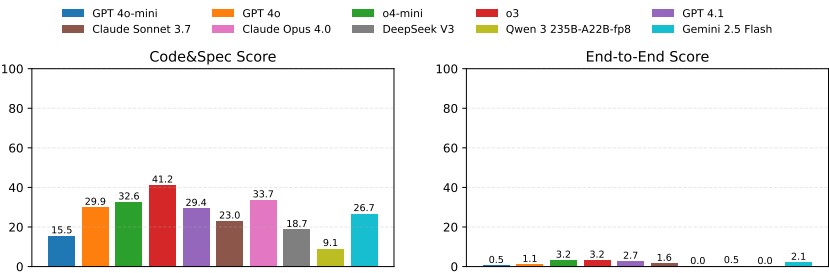

Figure 6: pass@1 performance of LLMs on VERINA's end-to-end verifiable code generation task.

**ProofGen is the major bottleneck for end-to-end verifiable code generation.** We further evaluate the models on the most challenging setting: end-to-end verifiable code generation, as defined in Section 4.2. We report the *Code&Spec Score*, where both generated code and specification should be correct, and the *End-to-End Score*, where additionally the proof verifying the generated code against the ground truth specification should be correct. As shown in Figure 6, simultaneously generating correct code and specifications is difficult, with the leading model, o3, achieving only 41.2%. Furthermore, the evaluation results confirm that ProofGen is the bottleneck in end-to-end verifiable code generation setting, with the leading model, o4-mini and o3, achieving only 3.2%.

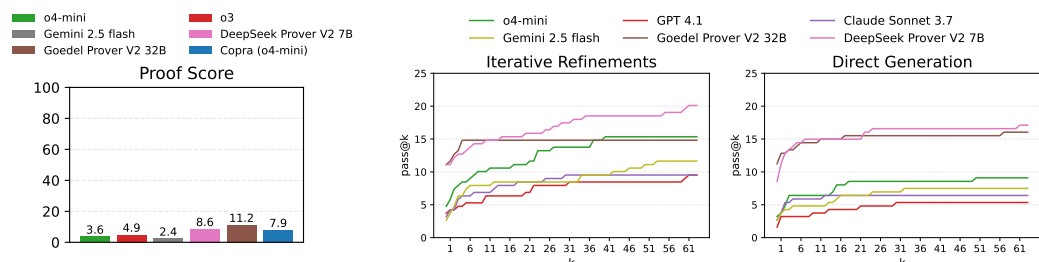

Figure 7: pass@1 for ProofGen across models and proving agent.

Figure 8: pass@$k$ performance of selective LLMs on ProofGen using proof refinement (left) and direct generation (right).

**Specialized provers and agentic methods improve proof success rate.** Given the limitations of general-purpose LLMs, we extend our evaluation to specialized theorem-proving models and agentic approaches. As shown in Figure 7, Goedel Prover V2 32B (Lin et al., 2025) and DeepSeek Prover V2 7B (Ren et al., 2025) achieve higher proof success rates compared to general-purpose models. We further evaluate Copra (Thakur et al., 2023), an agentic theorem-proving framework based on tree-search. We use o4-mini as the backbone model and allow at most 64 LLM queries for each sample. Copra demonstrates clear improvements over direct single-pass generation.

**Iterative proof refinement shows meaningful improvements.** For ProofGen task, besides pass@1, we also extend the evaluation of the four general-purpose models (o4-mini, GPT 4.1, Claude Sonnet 3.7, Gemini 2.5 Flash) alongside two specialized LLM-provers (Goedel Prover V2 32B (Lin et al., 2025) and DeepSeek Prover V2 7B (Ren et al., 2025)). We evaluate them with iterative proof refinement, where the evaluated model receives Lean verifier error messages and is prompted to revise its proof, and with direct generation, where the evaluated model generates responses independently without Lean feedback in each iteration. For all methods, we report pass@$k$, the success rate after $k$ rounds of iterations, for $k$ up 64. This metric investigates how much additional interaction helps repair the proof that a single-pass generation would miss, and whether providing Lean verifier feedback improves success rates compared to independent generation attempts.

As shown in Figure 8, iterative proof refinement reliably outperforms direct generation at matched query budgets on both general purpose and proof-specific models, underscoring the value of Lean verifier feedback. A detailed breakdown by problem difficulty is provided in Appendix D.

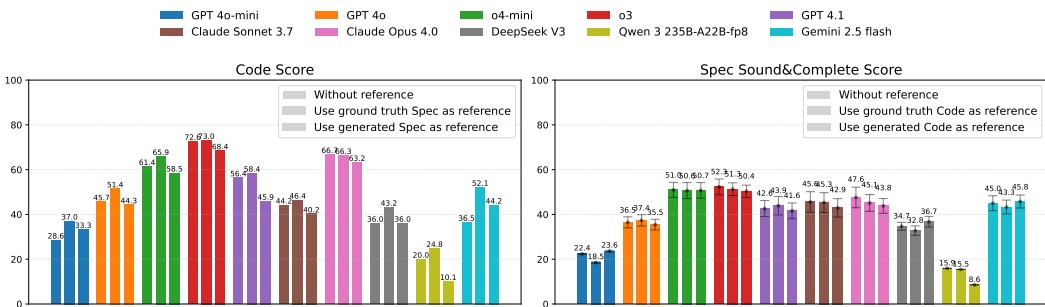

Figure 9: Impact of contextual information on CodeGen and SpecGen performance.

**Providing ground truth specification benefits CodeGen.** Providing ground truth specifications as context consistently improves CodeGen performance across models. Since the ground truth specifications cannot be used directly as code (as explained in Section 3.2), all CodeGen improvements rely on semantic understanding of the reference specification. On the contrary, providing ground truth code as context shows minimal or negative improvement for SpecGen. While it is possible for LLMs to directly use the ground truth code in the specification, manual inspection of our evaluation results reveals no evidence of such behaviors. This is likely because using code as specification is uncommon in standard development practices, and our prompts C.3 ask LLMs to focus on constraining code behavior rather than replicating implementation details. The asymmetry in using ground truth information for CodeGen versus SpecGen suggests that formal specifications effectively constrain and guide code synthesis, while verbose code implementations may introduce noise to or over-constrain specification generation rather than providing helpful guidance. Moreover, replacing ground truth with LLM-generated artifacts generally degrades performance, indicating that combined tasks are more challenging than individual tasks.

**Qualitative case studies.** We present detailed qualitative case studies with analysis of failure modes and success patterns across different tasks in Appendix E.

## 6 CONCLUSION AND DISCUSSION

We have introduced VERINA, a comprehensive benchmark comprising 189 carefully curated examples with detailed task descriptions, high-quality codes and specifications in Lean, and extensive test suites with full line coverage. This benchmark enables systematic assessment of various verifiable code generation capabilities, and our extensive evaluation result presents substantial challenges that expose limitations of state-of-the-art language models on verifiable code generation tasks. We hope that VERINA will serve as a valuable resource by providing both a rigorous evaluation framework and clear directions towards more reliable and formally verified automated programming systems.

**Limitations and future work.** Despite advancing the state-of-the-art in benchmarking verifiable code generation, VERINA has several limitations. First, its size (189 examples) is modest, scaling to a larger dataset suitable for finetuning likely requires automated annotation with LLM assistance. Second, it emphasizes simple, standalone coding problems, which is well-suited for benchmarking but not fully representative of complex real-world verification projects (Klein et al., 2009; Leroy et al., 2016). Our results demonstrate that current models struggle with VERINA, especially on ProofGen, with performance dropping substantially on harder instances (see Appendix D), indicating these fundamental capabilities must improve before tackling more difficult verification challenges. Third, while our current evaluation pipeline overcomes the limitation of current LLM theorem provers using comprehensive testing, the future advances in LLM theorem prover capabilities can enable stronger formal guarantees. Fourth, extending VERINA to ATP-based verification system like Dafny (Leino, 2010) or Verus (Lattuada et al., 2023) can strengthen VERINA's generalizability but requires significant effort, and we leave this as an important future work. Finally, while Lean programs in VERINA are newly written, the underlying task topics are drawn from widely used sources, posing a risk of data contamination. We provide detailed data contamination analysis and discussion in Appendix B.

## ACKNOWLEDGMENTS

This material is based upon work supported by the National Science Foundation under grant no. 2229876 and is supported in part by funds provided by the National Science Foundation, by the Department of Homeland Security, and by IBM. Any opinions, findings, and conclusions or recommendations expressed in this material are those of the author(s) and do not necessarily reflect the views of the National Science Foundation or its federal agency and industry partners.

## ETHICS STATEMENT

We adhere to the ICLR Code of Ethics and ensure compliance with all relevant dataset licenses, as detailed in Appendix C.1. All data used in this work are publicly available and collected strictly for academic research purposes with proper citation and attribution.

## REPRODUCIBILITY STATEMENT

We are committed to ensuring the reproducibility of our work. All code, benchmark datasets, and evaluation pipelines introduced in this paper are included in the supplementary materials, accompanied by detailed instructions for setup and usage. The dataset construction processes are described in Section 3.2. The evaluation metrics are described in Section 4. Additional implementation details and experimental settings are described in the appendix.

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

## A    EXTENDED DISCUSSION ON THE USE OF LEAN IN VERINA

This appendix provides extended discussion on the rationale for using Lean in VERINA, demonstrates its growing role in production code verification beyond mathematics, and explains how VERINA's findings transfer to broader verification ecosystems including automated theorem proving (ATP) systems like Dafny (Leino, 2010) and Verus (Lattuada et al., 2023).

**Lean in Production Code Verification.** While Lean originated in formalizing mathematics, recent years have witnessed Lean's substantial adoption for production code verification across diverse domains. Amazon Web Services (AWS) has invested heavily in Lean as verification infrastructure for critical production systems (de Moura). For example, AWS's Cedar project (Cutler et al., 2024) employs Lean to verify security properties of their policy language for cloud services authorization, handling authorization decisions for millions of resources. The LNSym (leanprover) project demonstrates Lean's capability in low-level verification by providing a symbolic simulator for Armv8 machine code, enabling verification at the hardware-software interface. Additionally, SampCert (de Medeiros et al., 2025b) represents a verified implementation of randomized algorithms deployed in production AWS services. The rise of Rust as a systems programming language has created demand for formal verification tools, and Lean has emerged as a viable platform. Aeneas (Ho & Protzenko, 2022) translates Rust programs into Lean for formal verification, enabling developers to prove properties about safe systems code. In blockchain, the Clear project (Nethermind) provides an interactive formal verification tool for Ethereum smart contracts using Lean, addressing the critical need for mathematical guarantees in high-stakes environments. Beyond direct production use, the CSLib project (Barrett et al.) represents a collaborative effort across academic institutions and industry partners to formalize undergraduate-level computer science in Lean, establishing reusable foundations for future verification projects. These diverse applications demonstrate that Lean's adoption extends well beyond mathematical formalization into practical software verification domains, validating its relevance as VERINA's platform.

**Transferable Insights Across Verification Paradigms.** While Lean's syntax differs from verification systems leveraging Automated Theorem Prover (ATP), the fundamental challenges in verifiable code generation are largely shared across verification paradigms. Both ITP and ATP frameworks require: (i) generating correct code and sound, complete specifications, and (ii) identifying key properties such as loop invariants for constructing proofs. VERINA's CodeGen and SpecGen tasks evaluate capabilities equally critical in ATP systems, which require the same semantic understanding regardless of surface syntax. For example, generating a sound and complete pre/post-conditions in Dafny requires the same semantic understanding as generating pre/post-conditions in Lean. The difficulty lies in specifying these properties correctly, not in the syntactic representation. For ProofGen, ATP systems automate proof search via SMT solvers but are not guaranteed to succeed on complex properties. When automation fails, LLM must generate additional guidance through assertions and annotations, which requires similar reasoning capabilities VERINA evaluates through explicit proof construction. Furthermore, Lean's dependent type system offers stronger expressiveness than the SMT-based specifications used in ATP systems, enabling verification of programs with higher-order functions and specifications. This greater expressiveness ensures that insights from VERINA generalize to ATP systems and beyond. These insights inform development of LLM-assisted code generation and verification workflow in both ITP and ATP paradigms, demonstrating that VERINA's findings extend beyond Lean-specific details to address fundamental challenges in verifiable code generation.

## B    EXTENDED DISCUSSION ON DATA CONTAMINATION ANALYSIS

VERINA draws algorithmic problems from popular sources, raising the possibility that models may have encountered similar problems during training and undermine the evaluation. To address this, we conducted systematic data contamination analysis and discuss how VERINA properly mitigates these risks.

**Direct contamination analysis.** We performed N-gram overlap analysis between VERINA's Lean ground truth solutions and the bigcode/the-stack pretraining dataset (Kocetkov et al., 2022), which contains approximately 550 million rows of coding files sourced from GitHub. Following standard decontamination practices like Qwen-2.5 Coder (Hui et al., 2024), we conducted 10-gram overlap

detection and found *zero* matches. This confirms that VERINA's Lean artifacts are novel and not present in public pretraining corpora, therefore no risk of direct contamination.

**Verification tasks differ fundamentally from simple code generation.** In verifiable code generation LLM are required to perform specification and proof generation beyond code generation, which are fundamentally different skills. Our evaluation shows that even for algorithmically familiar problems, the best LLMs struggle significantly with formal specification generation and proof generation, demonstrating that memorized algorithmic solutions do not transfer to verification tasks in current LLMs, effectively eliminating indirect contamination risks. The stark contrast between CodeGen and ProofGen success rates, despite both potentially benefiting from algorithmic familiarity, demonstrates that LLMs cannot formally reason about algorithms they may have seen and thus suggest a lack of deep understanding of the algorithm they use.

## C DATASETS AND DETAILED EXPERIMENTAL SETUP

### C.1 LICENSE

We ensure compliance with all relevant licenses: MBPP-DFY-50 (Misu et al., 2024) is licensed under GPL-3.0, while both CloverBench (Sun et al., 2024) and LiveCodeBench (Jain et al., 2025) use MIT licenses. Our datasets VERINA will be licensed under GPL-3.0. Consistent with established research practices (Hendrycks et al., 2021; Jain et al., 2025), we only use publicly available materials from competitive programming platforms such as LeetCode. Our collection and use of these problems is strictly for academic research purposes, and VERINA involves no model training or fine-tuning processes.

### C.2 MODEL CONFIGURATIONS AND COMPUTE

Table 3 presents the configuration details and total experiment costs for all twelve evaluated LLMs. For all LLMs, we use a temperature of 1.0 and a maximum output token budget of 10,000. For reasoning models, we use default settings of reasoning efforts or budgets. We host DeepSeek Prover V2 7B, Goedel Prover V2 32B, and Qwen 3 235B-A22B locally using 8 NVIDIA H100 80GB GPUs. We run other LLMs through APIs, for which we provide the total cost and cost per million tokens. The costs marked with asterisks include the additional expenses incurred during iterative proof refinement experiments, which required up to 64 refinement attempts per datapoint.

Table 3: Detailed configurations and costs for evaluated LLMs.

| Vendor | Model Name | Checkpoint | Type | Price ($/1M tokens) (Input / Output) | Cost |
|---|---|---|---|---|---|
| OpenAI | GPT 4o-mini | gpt-4o-mini-2024-07-18 | API | $0.15 / $0.60 | $10.94 |
| | GPT 4o | gpt-4o-2024-08-06 | API | $2.50 / $10.0 | $153.01 |
| | GPT 4.1 | gpt-4.1-2025-04-14 | API | $2.00 / $8.00 | $453.72[*] |
| | o4-mini | o4-mini-2025-04-16 | API | $1.10 / $4.40 | $894.38[*] |
| | o3 | o3-2025-04-16 | API | $2.00 / $8.00 | $121.70 |
| Anthropic | Claude Sonnet 3.7 | claude-3-7-sonnet-20250219 | API | $3.00 / $15.0 | $777.60[*] |
| | Claude Opus 4.0 | claude-opus-4-20250514 | API | $15.00 / $75.0 | $1197.39 |
| Google | Gemini 2.5 Flash | gemini-2.5-flash-preview-04-17 | API | $0.15 / $0.60 | $295.20[*] |
| DeepSeek | DeepSeek V3 | DeepSeek-V3-0324 | API | $1.25 / $1.25 | $51.15 |
| | DeepSeek Prover V2 7B | DeepSeek-Prover-V2-7B | GPU | - | - |
| Qwen | Qwen3 235B-A22B | Qwen3-235B-A22B-FP8 | GPU | - | - |
| Goedel-LM | Goedel Prover V2 32B | Goedel-Prover-V2-32B | GPU | - | - |

[*] Including costs for iterative proof refinement experiments.

## C.3 PROMPTS

We employ a consistent 2-shot prompting approach across all models and tasks to enhance output format adherence and task understanding. The 2-shot examples are excluded from the final benchmark evaluation. For each problem instance, we sample 5 responses from each model and calculate pass@1 metrics (Chen et al., 2021) using these 5 samples to ensure robust evaluation statistics. We utilize DSPy (Khattab et al., 2024) for structural prompting. We provide the detailed prompts in the following: Prompt 1 for CodeGen, Prompt 2 for SpecGen, Prompt 3 for ProofGen, and Prompt 4 for ProofGen with iterative refinement. For DeepSeek Prover V2 7B and Goedel Prover V2 32B, we used their own prompt templates for ProofGen to achieve optimal performance. Our control experiments revealed that using the standard DSPy prompts for these models resulted in a 0% success rate due to severe instruction-following failures. Specifically, they are not able to produce parsable output formats using the standard DSPy prompts.

---

**Prompt 1 (CodeGen)**

**Instructions**

```
You are an expert in Lean 4 programming and theorem proving.
Please generate a Lean 4 program that finishes the task described
    ↪ in
`task_description` using the template provided in `task_template`.
The `task_template` is a Lean 4 code snippet that contains
    ↪ placeholders
(warpped with {{}}) for the code to be generated.
The program should:
- Be well-documented with comments if necessary
- Follow Lean 4 best practices and use appropriate Lean 4 syntax
    ↪ and features
- DO NOT use Lean 3 syntax or features
- DO NOT import Std or Init
Hint:
- Use a[i]! instead of a[i] when a is an array or a list when
    ↪ necessary
```

- - - - - - - - - - - - - - - - - - - - - - - - - - - - - - - - - - - - - - - - - - - - -

**Input Fields**

- **task_description**
  ```
  Description of the Lean 4 programming task to be solved.
  ```

- **task_template**
  ```
  Lean 4 template with placeholders for code generation and optional
  reference specification.
  ```

- - - - - - - - - - - - - - - - - - - - - - - - - - - - - - - - - - - - - - - - - - - - -

**Output Fields**

- **imports**
  ```
  Imports needed for `code`. Keep it empty if not needed.
  ```

- **code_aux**
  ```
  Auxiliary definitions for `code`. Keep it empty if not needed.
  ```

- **code**
  ```
  Generated Lean 4 code following the template signature and complete
  the task.
  ```

**Prompt 2 (SpecGen)**

**Instructions**

```
You are an expert in Lean 4 programming and theorem proving.
Please generate a Lean 4 specification that constrains the program
implementation using the template provided in `task_template`.
The `task_template` is a Lean 4 code snippet that contains
    ↪ placeholders
(warpped with {{}}) for the spec to be generated.
The precondition should be as permissive as possible, and the
    ↪ postcondition
should model a sound an complete relationship between input and
    ↪ output of the
program based on the `task_description`.
The generated specification should:
- Be well-documented with comments if necessary
- Follow Lean 4 best practices and use appropriate Lean 4 syntax
    ↪ and features
- DO NOT use Lean 3 syntax or features
- DO NOT import Std or Init
- Only use `precond_aux` or `postcond_aux` when you cannot express
the precondition or postcondition in the main body of the
    ↪ specification
- add @[reducible, simp] attribute to the definitions in `
    ↪ precond_aux` or
`postcond_aux`
Hint:
- Use a[i]! instead of a[i] when a is an array or a list when
    ↪ necessary
```

**Input Fields**

- **task_description**
  ```
  Description of the Lean 4 programming task to be solved.
  ```

- **task_template**
  ```
  Lean 4 template with placeholders for specfication generation and
  optional reference code.
  ```

**Output Fields**

- **imports**
  ```
  Imports needed for `precond` and `postcond`. Keep it empty if not
  needed.
  ```

- **precond_aux**
  ```
  Auxiliary definitions for `precond`. Keep it empty if not needed.
  ```

- **precond**
  ```
  Generated Lean 4 code specifying the precondition.
  ```

- **postcond_aux**
  ```
  Auxiliary definitions for `postcond`. Keep it empty if not needed.
  ```

- **postcond**
  ```
  Generated Lean 4 code specifying the postcondition.
  ```

**Prompt 3 (ProofGen)**

**Instructions**

```
You are an expert in Lean 4 programming and theorem proving.
Please generate a Lean 4 proof that the program satisfies the
    ↪ specification
using the template provided in `task_template`.
The `task_template` is a Lean 4 code snippet that contains
    ↪ placeholders
(warpped with {{}}) for the proof to be generated.
The proof should:
- Be well-documented with comments if necessary
- Follow Lean 4 best practices and use appropriate Lean 4 syntax
    ↪ and features
- DO NOT use Lean 3 syntax or features
- DO NOT import Std or Init
- DO NOT use cheat codes like `sorry`
Hint:
- Unfold the implementation and specification definitions when
    ↪ necessary
- Unfold the precondition definitions at h_precond when necessary
```

**Input Fields**

- **task_description**
  Description of the Lean 4 programming task to be solved.

- **task_template**
  Lean 4 template with code and specification to be proved, and
  placeholders for proof generation.

**Output Fields**

- **imports**
  Imports needed for `proof`. Keep it empty if not needed.

- **proof_aux**
  Auxiliary definitions and lemma for `proof`. Keep it empty if not
  needed.

- **proof**
  Generated Lean 4 proof that the program satisfies the specification.

Prompt 4 (ProofGen with Iterative Refinement)

**Instructions**

```
You are an expert in Lean 4 programming and theorem proving.
Please generate a Lean 4 proof that the program satisfies the
    ↪ specification
using the template provided in `task_template`.
The `task_template` is a Lean 4 code snippet that contains
    ↪ placeholders
(warpped with {{}}) for the proof to be generated.
The proof should:
- Be well-documented with comments if necessary
- Follow Lean 4 best practices and use appropriate Lean 4 syntax
    ↪ and features
- DO NOT use Lean 3 syntax or features
- DO NOT import Std or Init
- DO NOT use cheat codes like `sorry`
Hint:
- Unfold the implementation and specification definitions when
    ↪ necessary
- Unfold the precondition definitions at h_precond when necessary

Furthermore, `prev_error` is the error message from the previous
    ↪ proving
attempt.
Please use the `prev_imports`, `prev_proof_aux`, and `prev_proof`
    ↪ as
references to improve the generated proof.
- You can ignore unused variable warnings in the error message.
```

------------------------------------------------------------

**Input Fields**

- **task_description**
  Description of the Lean 4 programming task to be solved.

- **task_template**
  Lean 4 template with code and specification to be proved, and
  placeholders for proof generation.

- **prev_imports**
  Previously generated imports for reference.

- **prev_proof_aux**
  Previously generated proof auxiliary for reference.

- **prev_proof**
  Previously generated proof for reference.

- **prev_error**
  Error message from the previous proving attempt.

------------------------------------------------------------

**Output Fields**

- **imports**
  Imports needed for `proof`. Keep it empty if not needed.

- **proof_aux**
  Auxiliary definitions and lemma for `proof`. Keep it empty if not
  needed.

- **proof**
  Generated Lean 4 proof that the program satisfies the specification.

## C.4 COMPARISON WITH CLEVER

As summarized in Table 4, CLEVER (Thakur et al., 2025) only supports evaluation of specification generation and specification-guided code generation. It lacks evaluation support for code generation, proof generation, specification inference from code, and fully end-to-end verifiable code generation. In contrast, VERINA fully covers all three foundational tasks and their flexible combinations, enabling a more comprehensive assessment of realistic verification workflows.

Moreover, CLEVER's SpecGen evaluation assumes access to a sound and complete ground truth specification for certification. However, if such ground truth specification is already available, there is little practical value in generating another, as developers would simply use the existing one. This reliance on ground truth specifications therefore limits CLEVER's applicability and prevents it from reflecting real-world scenarios. In contrast, VERINA employs a combined evaluation framework for specification (Section 4.1) leveraging both formal proving and comprehensive testing, which can reliably assess specification quality even when formal proofs are inconclusive.

Table 4: A detailed comparison of VERINA with the concurrent work CLEVER (Thakur et al., 2025) on supported tasks in verifiable code generation. ● means fully supported, ○ means unsupported.

| | Foundational Tasks (Section 4.1) | | | Task Combinations (Section 4.2) | | |
|---|---|---|---|---|---|---|
| | **CodeGen**
(Desc → Code) | **SpecGen**
(Desc → Spec) | **ProofGen**
(Code+Spec → Proof) | **Specification-Guided Code Generation**
(Desc + Spec → Code + Proof) | **Specification Inference From Code**
(Desc + Code → Spec + Proof) | **End-to-End Verifiable Code Generation**
(Desc → Code + Spec + Proof) |
| CLEVER (Thakur et al., 2025) | ○ | ● | ○ | ● | ○ | ○ |
| VERINA | ● | ● | ● | ● | ● | ● |

## C.5 Implementation of Evaluation Metrics in Lean

In Section 4.1, we provide a high-level description of our evaluation metrics for the three foundational tasks of verifiable code generation. Now we describe how we implement these metrics in Lean 4.

**Proof evaluation.** We directly evaluate generated proofs using the Lean compiler and filter out any proofs containing placeholders, as described in Section 4.1.

**Code evaluation.** We evaluate generated code on unit tests using `#guard` statements in Lean 4, ensuring the implementation produces correct outputs for given inputs. The evaluation harness for generated codes is illustrated in Figure 10.

```
1   import Mathlib
2   import Plausible
3
4   -- Definitions for code (removeElement) omitted for brevity
5
6   -- Evaluate code correctness using positive test cases
7   #guard removeElement (#[1, 2, 3, 4, 5]) (2) (by sorry) == (#[1, 2, 4, 5]) -- Should pass
```

Figure 10: Example (`verina_basic_29`): Evaluating the correctness of LLM-generated code using unit tests in Lean 4.

**Specification evaluation.** Recall in Section 4.1, we define the soundness and completeness of model-generated pre-condition $\hat{P}$ and post-condition $\hat{Q}$ in relation to their ground truth counterparts $P$ and $Q$: (i) $\hat{P}$ is sound iff $\forall \overline{x}.P(\overline{x}) \Rightarrow \hat{P}(\overline{x})$; (ii) $\hat{P}$ is complete iff $\forall \overline{x}.\hat{P}(\overline{x}) \Rightarrow P(\overline{x})$; (iii) $\hat{Q}$ is sound iff $\forall \overline{x}, y.P(\overline{x}) \wedge \hat{Q}(\overline{x}, y) \Rightarrow Q(\overline{x}, y)$; (iv) $\hat{Q}$ is complete iff $\forall \overline{x}, y.P(\overline{x}) \wedge Q(\overline{x}, y) \Rightarrow \hat{Q}(\overline{x}, y)$.

Our specification evaluation pipeline first attempts to establish the soundness and completeness of generated specifications against the ground truth using LLM-based provers. When the proving step is inconclusive, the evaluator proceeds to testing, where we only require that $\overline{x}$ and $y$ are from our test suite. Our quality assurance process in Section 3.2 ensures that all ground truth pre-conditions and post-conditions pass our positive tests and do not pass our negative tests. Therefore, we can simplify the soundness and completeness metrics as follows:

- Deciding the soundness of $\hat{P}$ is equivalent to verifying whether $\hat{P}(\overline{x})$ holds for all positive tests $\overline{x}$ in our test suite. This is because for all negative tests $\overline{x}$, $P(\overline{x})$ does not hold, making $P(\overline{x}) \Rightarrow \hat{P}(\overline{x})$ true by default. For all positive tests $\overline{x}$, $P(\overline{x})$ holds, and $P(\overline{x}) \Rightarrow \hat{P}(\overline{x})$ is true iff $\hat{P}(\overline{x})$ is true.

- Similarly, deciding the completeness of $\hat{P}$ is equivalent to verifying whether $\hat{P}(\overline{x})$ does not hold for all negative tests $\overline{x}$ in our test suite.

- The soundness of $\hat{Q}$ can be evaluated using our negative test cases.

- The completeness of $\hat{Q}$ can be evaluated using our positive test cases.

For each test case evaluation, we employ the two-step approach described in Section 4.1. First, we check if the relationship (with the specific test case incorporated) is directly decidable in Lean 4 on the test case via `decide`. If not, we proceed to property-based testing using `plausible` tactic. The evaluation implementation in Lean 4 is illustrated in Figures 11 and 12.

To further examine the role of proofs within our evaluation pipeline, we analyze how often LLM-based provers succeed in establishing the soundness and completeness of generated specifications against the ground truth. In this setup, we use o4-mini and Claude Sonnet 3.7 to construct Lean proofs for the required logical relationships and compare the results with the testing-based evaluation results. Table 5 summarizes the outcomes. Proof success rates are very low, below 4% across all cases, while testing recognizes more than 40% of generated specifications as sound and complete. We have examined all specifications marked as sound and complete by formal proofs. We observe that whenever proofs succeed they always agree with testing, confirming their validity. However, when proofs fail but testing reports correctness, manual inspection of 20 randomly selected disagreements shows that the testing outcome is always correct.

These results indicate that while proofs provide the formal guarantees of the evaluation results when they succeed, current LLM provers are incapable of serving as a reliable metric with high inconclusive

rates. Testing-based evaluation methods achieve high empirical accuracy and reliably identify sound and complete specifications even when proofs are inconclusive and therefore play an important role in ensuring robust and comprehensive specification evaluation when the proving-based evaluation is inconclusive. LLMs' proof capabilities and stability are rapidly improving with newer prover models, stronger proof search agents, and new automation tactics like `grind`. This will make our SpecGen metric increasingly powerful over time.

We further analyze the sensitivity of our SpecGen evaluation (on o4-mini results) to the property-based testing budget, varying the number of generated test instances from 10 to 2,000 across 5 random seeds. Table 6 demonstrates that VERINA's choice of 1,000 test instances is sufficient, as increasing the budget to 2,000 instances provides minimal additional benefit, and the standard deviations across seeds are small. All evaluation components contribute meaningfully to the final determination, demonstrating the comprehensiveness of our specification evaluation approach. The most variable component (property-based testing) contributes less than 13% of cases. As LLM-based theorem provers continue to improve, we expect the "Guaranteed to Hold (proved)" percentage to increase, providing more formal guarantees to the SpecGen evaluation.

Table 5: Evaluation of generated specifications for soundness and completeness. Rows indicate the model that generated the specification, while columns indicate the prover used to check correctness. The last column shows results from our testing-based evaluation.

| Spec generated by | Proved sound and complete by (%) | | Sound and complete by testing (%) |
|---|---|---|---|
| | o4-mini | Claude Sonnet 3.7 | |
| o4-mini | 3.7 | 1.6 | 51.0 |
| Claude Sonnet 3.7 | 3.7 | 2.6 | 41.6 |

Table 6: Sensitivity analysis of SpecGen evaluation results (o4-mini) with varying property-based testing budgets. Results are averaged over 5 random seeds. The "Unknown" column includes the standard deviation across seeds.

| # Test Budget | Guaranteed to Hold (Proved) (%) | Might Hold (%) | | Guaranteed to Not Hold (Counterexample) (%) | Unknown (%) | Cannot Compile (%) |
|---|---|---|---|---|---|---|
| | | Unit Tests | PBT | | | |
| 10 | 3.7 | 38.7 | 2.1 | 21.0 | $10.5 \pm 0.41$ | 24.1 |
| 100 | 3.7 | 38.7 | 5.6 | 21.3 | $7.3 \pm 0.59$ | 24.1 |
| 1000 | 3.7 | 38.7 | 5.7 | 21.3 | $7.2 \pm 0.13$ | 24.1 |
| 2000 | 3.7 | 38.7 | 5.7 | 21.3 | $7.2 \pm 0.11$ | 24.1 |

```
1    import Mathlib
2    import Plausible
3
4    -- Definitions for pre-condition (removeElement_precond) omitted for brevity
5
6    -- Evaluate precond soundness with positive test cases
7    #guard decide (removeElement_precond (#[1, 2, 3, 4, 5]) (2))
8    example : (removeElement_precond (#[1, 2, 3, 4, 5]) (2)) := by -- Should pass
9        unfold removeElement_precond
10       simp_all! (config := { failIfUnchanged := false })
11       simp (config := { failIfUnchanged := false }) [*]
12       plausible (config := { numInst := 1000, maxSize := 100, numRetries := 20, randomSeed := some 42})
13   example : ¬(removeElement_precond (#[1, 2, 3, 4, 5]) (2)) := by -- Should fail
14       unfold removeElement_precond
15       simp_all! (config := { failIfUnchanged := false })
16       simp (config := { failIfUnchanged := false }) [*]
17       plausible (config := { numInst := 1000, maxSize := 100, numRetries := 20, randomSeed := some 42})
18
19   -- Evaluate precond completeness with negative test cases
20   #guard decide (¬ (removeElement_precond (#[1]) (2)))
21   example : ¬(removeElement_precond (#[1]) (2)) := by -- Should pass
22       unfold removeElement_precond
23       simp_all! (config := { failIfUnchanged := false })
24       simp (config := { failIfUnchanged := false }) [*]
25       plausible (config := { numInst := 1000, maxSize := 100, numRetries := 20, randomSeed := some 42})
26   example : (removeElement_precond (#[1]) (2)) := by -- Should fail
27       unfold removeElement_precond
28       simp_all! (config := { failIfUnchanged := false })
29       simp (config := { failIfUnchanged := false }) [*]
30       plausible (config := { numInst := 1000, maxSize := 100, numRetries := 20, randomSeed := some 42})
```

Figure 11: Example (verina_basic_29): Evaluating pre-condition soundness and completeness using unit tests in Lean 4.

```
1    import Mathlib
2    import Plausible
3
4    -- Definitions for post-condition (removeElement_postcond) omitted for brevity
5
6    -- Evaluate postcond completeness with positive test cases
7    #guard decide (removeElement_postcond (#[1, 2, 3, 4, 5]) (2) (#[1, 2, 4, 5]) (by sorry))
8    example : (removeElement_postcond (#[1, 2, 3, 4, 5]) (2) (#[1, 2, 4, 5]) (by sorry)) := by -- Should pass
9        unfold removeElement_postcond
10       simp_all! (config := { failIfUnchanged := false })
11       simp (config := { failIfUnchanged := false }) [*]
12       plausible (config := { numInst := 1000, maxSize := 100, numRetries := 20, randomSeed := some 42})
13   example : ¬(removeElement_postcond (#[1, 2, 3, 4, 5]) (2) (#[1, 2, 4, 5]) (by sorry)) := by -- Should fail
14       unfold removeElement_postcond
15       simp_all! (config := { failIfUnchanged := false })
16       simp (config := { failIfUnchanged := false }) [*]
17       plausible (config := { numInst := 1000, maxSize := 100, numRetries := 20, randomSeed := some 42})
18
19   -- Evaluate postcond soundness with negative test cases
20   #guard decide (¬ (removeElement_postcond (#[1, 2, 3, 4, 5]) (2) (#[1, 2, 3, 5]) (by sorry)))
21   example : ¬(removeElement_postcond (#[1, 2, 3, 4, 5]) (2) (#[1, 2, 3, 5]) (by sorry)) := by -- Should pass
22       unfold removeElement_postcond
23       simp_all! (config := { failIfUnchanged := false })
24       simp (config := { failIfUnchanged := false }) [*]
25       plausible (config := { numInst := 1000, maxSize := 100, numRetries := 20, randomSeed := some 42})
26   example : (removeElement_postcond (#[1, 2, 3, 4, 5]) (2) (#[1, 2, 3, 5]) (by sorry)) := by -- Should fail
27       unfold removeElement_postcond
28       simp_all! (config := { failIfUnchanged := false })
29       simp (config := { failIfUnchanged := false }) [*]
30       plausible (config := { numInst := 1000, maxSize := 100, numRetries := 20, randomSeed := some 42})
```

Figure 12: Example (verina_basic_29): Evaluating post-condition soundness and completeness using unit tests in Lean 4.

# D    ADDITIONAL EXPERIMENTAL EVALUATION RESULTS

Based on the construction methodology of VERINA datasets in Section 3.2, we categorize the problems translated from human-written Dafny datasets as VERINA-A and the problems written from scratch as VERINA-B.

**VERINA-B is much more challenging than VERINA-A.** The comparison between VERINA-A and VERINA-B in Figure 13 reveals substantial difficulty gaps on all three tasks. This demonstrates that problem complexity significantly impacts all aspects of verifiable code generation, and VERINA-B provides a valuable challenge for advancing future research in this domain.

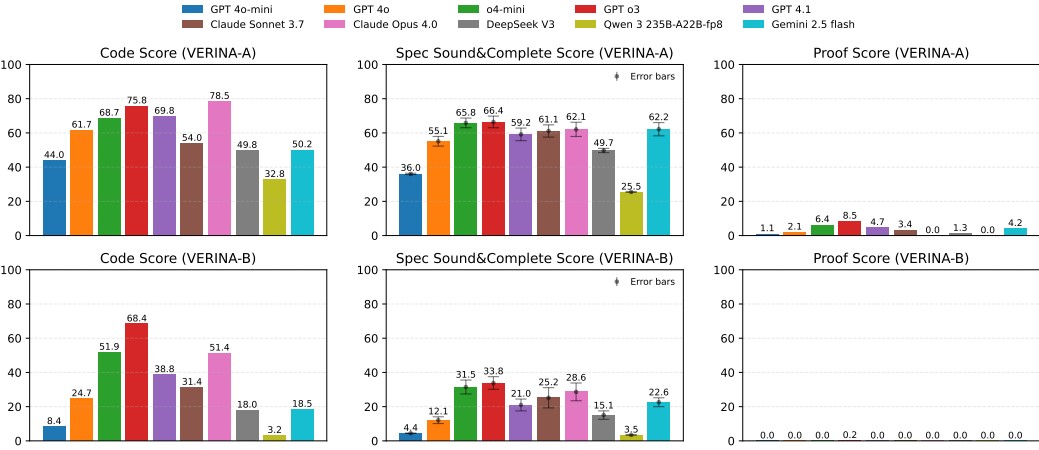

Figure 13: pass@1 performance on three foundational tasks for VERINA-A and VERINA-B.

**Achieving simultaneous soundness and completeness poses great challenge, particularly for post-conditions.** As shown in Figure 14, the substantial performance gap between preconditions and postconditions confirms that generating complex input-output relationships remains significantly more challenging than input validation constraints. Furthermore, the drop in performance when requiring both soundness and completeness simultaneously—compared to achieving either individually—demonstrates that partial correctness is insufficient and justifies our comprehensive evaluation framework for specification quality.

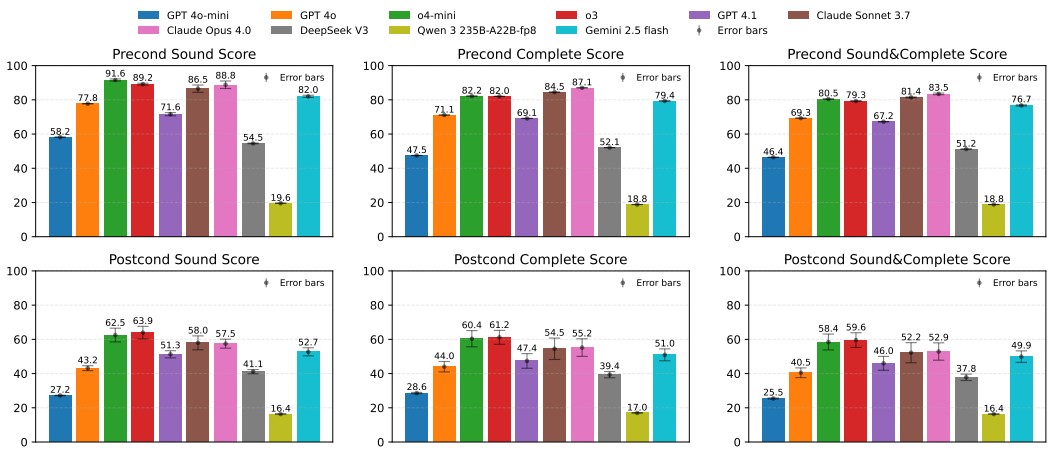

Figure 14: Detailed performance of LLMs on VERINA's SpecGen task.

**Code summarization does not consistently benefit SpecGen.** We conducted an ablation study where we replaced the full code reference with behavior-only summaries (generated by prompting LLMs to extract high-level contracts and behavior descriptions without implementation details). The results in Table 7 show highly model-dependent effects: o4-mini improves slightly with summaries (51.0%

$\rightarrow$ 53.2%), GPT 4.1 shows smaller difference ($\sim$42-44%), while Gemini 2.5 Flash performance is hurt by summaries (45.0% $\rightarrow$ 37.3%). We note that LLM-generated code summaries can themselves be detail-heavy and potentially more confusing than the original code, especially when attempting to describe low-level implementation logic in natural language, which may explain why summaries do not consistently improve performance and can even hurt it.

Table 7: Ablation study on SpecGen performance using code summaries versus full code references. "Ref" indicates the reference provided to the model; "GT" is Ground Truth, "Gen" is Generated.

| Model | No Ref (%) | GT Code (%) | Gen Code (%) | GT Summary (%) | Gen Summary (%) |
|---|---|---|---|---|---|
| o4-mini | 51.0 | 50.6 | 50.7 | 53.2 | 52.6 |
| GPT 4.1 | 42.6 | 43.9 | 41.6 | 43.9 | 41.8 |
| Gemini 2.5 Flash | 45.0 | 43.3 | 45.8 | 37.3 | 37.4 |

**Naive proof refinement gains diminish when problem is difficult.** As shown in Figure 15, iterative proof refinement yields substantial improvements on simpler problems but only modest gains on more complex ones. For example, o4-mini improves from 7.41% to 22.22% on VERINA-A after 64 iterations, while on VERINA-B the success rate rises only from 1.23% to 6.17%. Specialized provers like Goedel Prover V2 and DeepSeek Prover V2 generally outperform general-purpose models, yet o4-mini remains surprisingly competitive on difficult instances, achieving stronger iterative refinement gains on VERINA-B. This suggests that while verifier feedback is crucial, naive refinement strategies struggle to overcome the inherent complexity of challenging proofs, and that general-purpose LLMs can still contribute meaningfully in difficult settings.

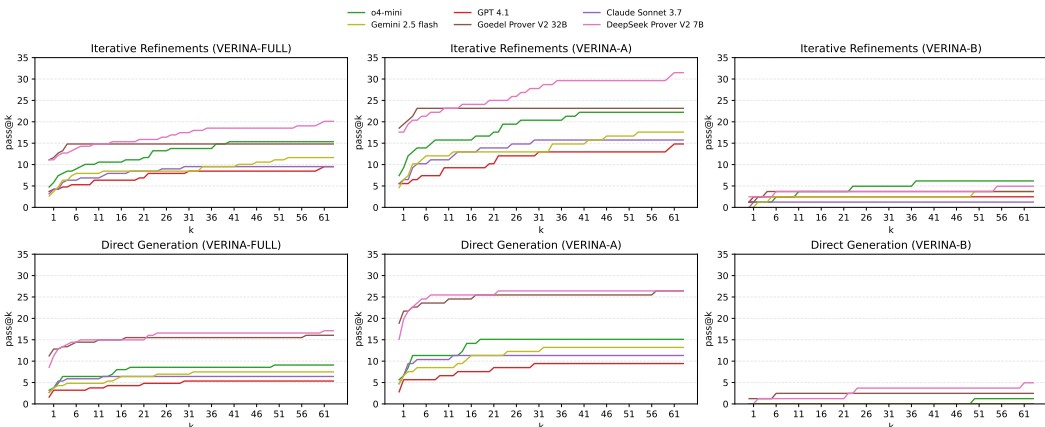

Figure 15: Breakdown of iterative refinement versus direct generation across different subsets. Refinement yields large gains on VERINA-A but limited improvements on VERINA-B.

**Iterative refinement increases proof verbosity.** We manually inspected and analyzed the structural differences between the 46 human-written ground truth proofs and successful proofs generated by models. As shown in Figure 16, human proofs are highly concise, averaging 169.6 characters, as experts effectively utilize automation tactics (e.g., `simp`, `aesop`) and standard library lemmas. In contrast, while single-pass LLM generation produces similarly short proofs, iterative refinement leads to a monotonic increase in verbosity, with the highest model (Gemini 2.5 Flash) reaching $> 1200$ characters after 64 iterations. Moreover, manual inspection reveals that LLMs often cannot correctly identify or use relevant lemmas from the standard library, leading them to explicitly prove tedious intermediate goals that humans would automate. This suggests improving LLMs' understanding of proof automation and lemma usage is a critical direction for future work.

**Iterative refinement is more cost-effective than COPRA.** We conducted a budget-normalized analysis to compare the marginal utility of iterative refinement against the agentic COPRA framework using o4-mini (Figure 17). Iterative refinement proves significantly more cost-effective, achieving an 8.99% overall success rate at a 50k token budget compared to COPRA's 4.76%. With the budget extended to 350k tokens, iterative refinement scales to 14.29% while Copra saturates at 7.94%,

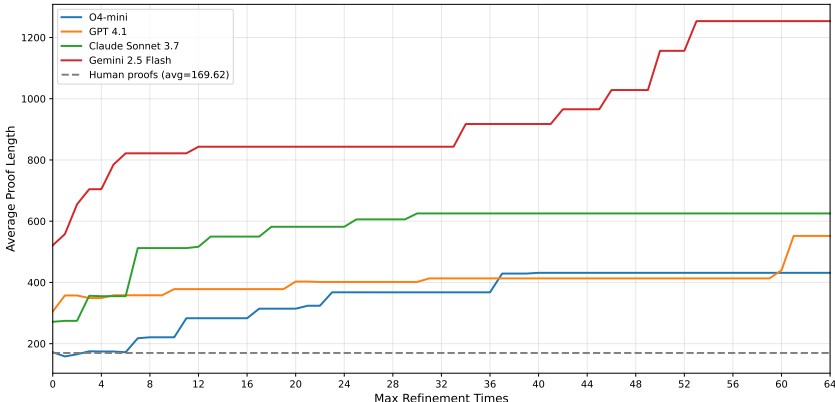

Figure 16: Comparison of average proof length between human-written ground truth and successful proofs.

with a substantially lower average cost per successful proof (57,594 tokens vs. 84,027). Stratified analysis reveals that returns diminish sharply with problem difficulty: while iterative refinement achieves steady gains on VERINA-A (reaching 22.22%), performance on the harder VERINA-B subset saturated early at 3.70% (compared to Copra's 1.23%), indicating that increased inference budgets alone cannot overcome fundamental reasoning gaps in complex verification tasks.

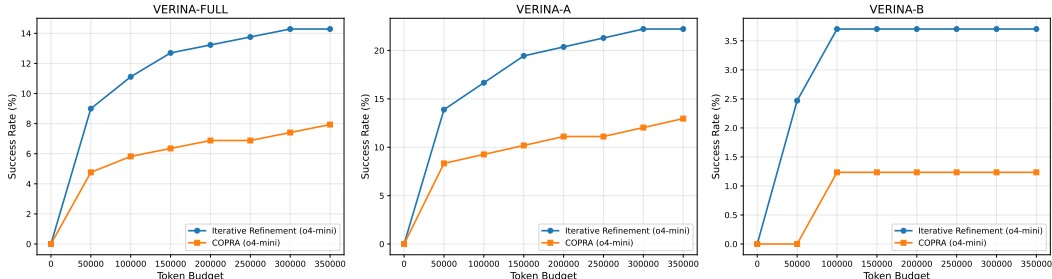

Figure 17: Budget-normalized comparison of proof success rates between iterative refinement and COPRA (using o4-mini) on the ProofGen task. Iterative refinement demonstrates higher marginal utility across all token budgets compared to the agentic COPRA framework, though performance gains on the harder VERINA-B subset remain limited for both approaches.

**Detailed performance breakdown.** Tables 8 to 16 provide detailed breakdowns of model performance across the three foundational tasks. They reveal that syntax incorrectness and use of non-existent library functions (as demonstrated in Appendix E) represent the major problems, especially for less capable models. Specifically, after manual inspection of the evaluation result, Qwen 3 235B-A22B-FP8 suffers from instruction following ability, failing to output the desired format specified in our prompts (cf. Appendix C.3). The relatively low unknown percentages across most evaluations demonstrate that our specification evaluation metric is reliable. Pre-conditions are generally simpler than post-conditions, resulting in lower unknown rates during evaluation. More capable models often generate specifications with more complicated logical structures, leading to higher unknown percentages in post-condition evaluation. We present a case study in Appendix E on the challenge of automatically evaluating LLM-generated specifications. In our main results, we report the uncertainty from unknown cases using error bars, where the lower bound represents the Pass% in the table and the upper bound represents Pass%+Unknown% in the table.

**ProofGen failure analysis.** To systematically diagnose bottlenecks in proof generation, we categorized failure modes across four top-performing models up to 64 refinements, as detailed in Tables 17 to 19. Our analysis reveals distinct failure signatures: Gemini 2.5 Flash and GPT 4.1 primarily struggle with *Incomplete Proofs* (77.55% and 38.24% respectively), often leaving goals unsolved after exhausting initial tactics. In contrast, Claude Sonnet 3.7 and o4-mini frequently resort to *Cheat*

*Codes* (e.g., `sorry`), which account for 48.53% and 28.30% of their failures, suggesting a tendency to explicitly acknowledge their inability to complete the proof. Additionally, errors stemming from *Unknown Identifiers, Tactics, and Constants* consistently account for 15–30% of failures across all models, underscoring a pervasive lack of familiarity with the specific syntax and standard library of the Lean ecosystem.

Table 8: Detailed performance of CodeGen.

| Model | Cannot Compile% | Fail Unit Test% | Pass% |
|---|---|---|---|
| GPT 4o-mini | 70.1 | 1.4 | 28.6 |
| GPT 4o | 51.6 | 2.8 | 45.7 |
| GPT 4.1 | 40.5 | 3.1 | 56.4 |
| o4-mini | 34.1 | 4.5 | 61.4 |
| o3 | 25.8 | 1.6 | 72.6 |
| Claude Sonnet 3.7 | 54.1 | 1.7 | 44.2 |
| Claude Opus 4.0 | 30.6 | 2.7 | 66.7 |
| Gemini 2.5 Flash | 62.9 | 0.6 | 36.5 |
| DeepSeek V3 | 62.3 | 1.7 | 36.0 |
| Qwen 3 235B-A22B-FP8 | 80.0 | 0.0 | 20.0 |

Table 9: Detailed performance of CodeGen on VERINA-A.

| Model | Cannot Compile% | Fail Unit Test% | Pass% |
|---|---|---|---|
| GPT 4o-mini | 54.9 | 1.1 | 44.0 |
| GPT 4o | 35.7 | 2.6 | 61.7 |
| GPT 4.1 | 27.4 | 2.8 | 69.8 |
| o4-mini | 28.3 | 3.0 | 68.7 |
| o3 | 22.8 | 1.3 | 75.9 |
| Claude Sonnet 3.7 | 45.7 | 0.4 | 54.0 |
| Claude Opus 4.0 | 21.1 | 0.4 | 78.5 |
| Gemini 2.5 Flash | 49.3 | 0.6 | 50.2 |
| DeepSeek V3 | 48.7 | 1.5 | 49.8 |
| Qwen3 235B-A22B-FP8 | 67.2 | 0.0 | 32.8 |

Table 10: Detailed performance of CodeGen on VERINA-B.

| Model | Cannot Compile% | Fail Unit Test% | Pass% |
|---|---|---|---|
| GPT 4o-mini | 89.9 | 1.7 | 8.4 |
| GPT 4o | 72.4 | 3.0 | 24.7 |
| GPT 4.1 | 57.8 | 3.5 | 38.8 |
| o4-mini | 41.7 | 6.4 | 51.9 |
| o3 | 29.6 | 2.0 | 68.4 |
| Claude Sonnet 3.7 | 65.2 | 3.5 | 31.4 |
| Claude Opus 4.0 | 43.0 | 5.7 | 51.4 |
| Gemini 2.5 Flash | 80.7 | 0.7 | 18.5 |
| DeepSeek V3 | 80.0 | 2.0 | 18.0 |
| Qwen3 235B-A22B-FP8 | 96.8 | 0.0 | 3.2 |

Table 11: Detailed performance of SpecGen for pre-condition.

| Model | Cannot Compile% | Soundness | | | Completeness | | |
|---|---|---|---|---|---|---|---|
| | | Pass% | Fail% | Unknown% | Pass% | Fail% | Unknown% |
| GPT 4o-mini | 40.8 | 58.2 | 1.1 | 0.0 | 47.5 | 11.8 | 0.0 |
| GPT 4o | 19.8 | 77.7 | 1.8 | 0.8 | 71.1 | 8.7 | 0.4 |
| GPT 4.1 | 24.3 | 70.7 | 1.1 | 4.0 | 69.1 | 3.5 | 3.1 |
| o4-mini | 5.4 | 91.0 | 0.6 | 3.0 | 82.1 | 10.7 | 1.8 |
| o3 | 6.2 | 88.7 | 1.6 | 3.5 | 82.0 | 9.6 | 2.1 |
| Claude Sonnet 3.7 | 4.9 | 84.4 | 2.3 | 8.5 | 84.5 | 3.7 | 6.8 |
| Claude Opus 4.0 | 4.5 | 86.6 | 1.7 | 7.2 | 87.1 | 3.3 | 5.1 |
| Gemini 2.5 Flash | 14.7 | 81.4 | 1.5 | 2.5 | 79.4 | 5.0 | 1.0 |
| DeepSeek V3 | 43.7 | 54.3 | 0.8 | 1.2 | 52.1 | 3.1 | 1.1 |
| Qwen 3 235B-A22B-FP8 | 80.4 | 19.6 | 0.0 | 0.0 | 18.8 | 0.8 | 0.0 |

Table 12: Detailed performance of SpecGen for pre-condition on VERINA-A.

| Model | Cannot Compile% | Soundness | | | Completeness | | |
|---|---|---|---|---|---|---|---|
| | | Pass% | Fail% | Unknown% | Pass% | Fail% | Unknown% |
| GPT 4o-mini | 20.8 | 78.1 | 1.1 | 0.0 | 65.1 | 14.2 | 0.0 |
| GPT 4o | 10.8 | 88.1 | 0.6 | 0.6 | 82.5 | 6.6 | 0.2 |
| GPT 4.1 | 9.8 | 85.7 | 0.8 | 3.8 | 82.3 | 4.7 | 3.2 |
| o3 | 3.2 | 93.4 | 0.9 | 2.5 | 87.2 | 7.6 | 2.1 |
| o4-mini | 4.0 | 92.1 | 0.9 | 3.0 | 88.1 | 6.4 | 1.5 |
| Claude Sonnet 3.7 | 0.0 | 93.4 | 0.9 | 5.7 | 90.6 | 4.7 | 4.7 |
| Claude Opus 4.0 | 1.3 | 92.1 | 0.9 | 5.7 | 90.9 | 4.7 | 3.0 |
| Gemini 2.5 Flash | 5.5 | 91.5 | 0.8 | 2.3 | 88.9 | 5.3 | 0.4 |
| DeepSeek V3 | 26.4 | 71.1 | 0.8 | 1.7 | 67.9 | 4.0 | 1.7 |
| Qwen3 235B-A22B-FP8 | 69.4 | 30.6 | 0.0 | 0.0 | 29.4 | 1.1 | 0.0 |

Table 13: Detailed performance of SpecGen for pre-condition on VERINA-B.

| Model | Cannot Compile% | Soundness | | | Completeness | | |
|---|---|---|---|---|---|---|---|
| | | Pass% | Fail% | Unknown% | Pass% | Fail% | Unknown% |
| GPT 4o-mini | 66.9 | 32.1 | 1.0 | 0.0 | 24.4 | 8.6 | 0.0 |
| GPT 4o | 31.6 | 64.0 | 3.5 | 1.0 | 56.3 | 11.4 | 0.7 |
| GPT 4.1 | 43.2 | 51.1 | 1.5 | 4.2 | 51.9 | 2.0 | 3.0 |
| o4-mini | 7.2 | 89.6 | 0.3 | 3.0 | 74.3 | 16.3 | 2.2 |
| o3 | 10.1 | 82.5 | 2.5 | 4.9 | 75.3 | 12.4 | 2.2 |
| Claude Sonnet 3.7 | 11.4 | 72.6 | 4.0 | 12.1 | 76.5 | 2.5 | 9.6 |
| Claude Opus 4.0 | 8.6 | 79.5 | 2.7 | 9.1 | 82.0 | 1.5 | 7.9 |
| Gemini 2.5 Flash | 26.7 | 68.2 | 2.5 | 2.7 | 66.9 | 4.7 | 1.7 |
| DeepSeek V3 | 66.4 | 32.4 | 0.7 | 0.5 | 31.4 | 2.0 | 0.3 |
| Qwen3 235B-A22B-FP8 | 94.8 | 5.2 | 0.0 | 0.0 | 4.9 | 0.3 | 0.0 |

Table 14: Detailed performance of SpecGen for post-condition.

| Model | Cannot Compile% | Soundness | | | Completeness | | |
|---|---|---|---|---|---|---|---|
| | | Pass% | Fail% | Unknown% | Pass% | Fail% | Unknown% |
| GPT 4o-mini | 68.3 | 27.1 | 4.2 | 0.4 | 28.2 | 2.6 | 0.9 |
| GPT 4o | 49.1 | 41.7 | 4.6 | 4.6 | 41.0 | 1.8 | 8.1 |
| GPT 4.1 | 41.8 | 49.2 | 1.8 | 7.2 | 43.1 | 0.8 | 14.3 |
| o4-mini | 22.7 | 58.5 | 3.1 | 15.7 | 55.6 | 2.7 | 19.0 |
| o3 | 23.1 | 60.2 | 1.8 | 14.9 | 57.1 | 2.3 | 17.5 |
| Claude Sonnet 3.7 | 30.6 | 53.9 | 3.2 | 12.3 | 48.2 | 1.6 | 19.6 |
| Claude Opus 4.0 | 27.3 | 54.9 | 2.5 | 15.4 | 50.1 | 1.4 | 21.3 |
| Gemini 2.5 Flash | 40.6 | 50.4 | 1.5 | 7.5 | 47.5 | 1.0 | 10.9 |
| DeepSeek V3 | 53.9 | 39.9 | 2.6 | 3.6 | 37.5 | 3.6 | 4.9 |
| Qwen 3 235B-A22B-FP8 | 83.0 | 16.4 | 0.6 | 0.0 | 17.0 | 0.0 | 0.0 |

Table 15: Detailed performance of SpecGen for post-condition on VERINA-A.

| Model | Cannot Compile% | Soundness | | | Completeness | | |
|---|---|---|---|---|---|---|---|
| | | Pass% | Fail% | Unknown% | Pass% | Fail% | Unknown% |
| GPT 4o-mini | 51.5 | 41.9 | 5.9 | 0.8 | 44.9 | 2.3 | 1.3 |
| GPT 4o | 30.8 | 61.3 | 4.3 | 3.6 | 60.6 | 1.5 | 7.2 |
| GPT 4.1 | 27.4 | 65.9 | 1.9 | 4.9 | 60.4 | 0.8 | 11.5 |
| o4-mini | 16.8 | 73.8 | 1.3 | 8.1 | 70.0 | 0.8 | 12.5 |
| o3 | 14.3 | 73.4 | 0.9 | 11.3 | 70.6 | 1.3 | 13.8 |
| Claude Sonnet 3.7 | 22.6 | 68.1 | 2.5 | 6.8 | 64.0 | 1.1 | 12.3 |
| Claude Opus 4.0 | 19.4 | 69.4 | 1.9 | 9.3 | 65.7 | 0.2 | 14.7 |
| Gemini 2.5 Flash | 24.0 | 69.4 | 1.1 | 5.5 | 63.6 | 0.8 | 11.7 |
| DeepSeek V3 | 39.4 | 56.6 | 1.5 | 2.5 | 54.0 | 2.5 | 4.2 |
| Qwen3 235B-A22B-FP8 | 72.8 | 26.2 | 0.9 | 0.0 | 27.2 | 0.0 | 0.0 |

Table 16: Detailed performance of SpecGen for post-condition on VERINA-B.

| Model | Cannot Compile% | Soundness | | | Completeness | | |
|---|---|---|---|---|---|---|---|
| | | Pass% | Fail% | Unknown% | Pass% | Fail% | Unknown% |
| GPT 4o-mini | 90.4 | 7.7 | 2.0 | 0.0 | 6.4 | 3.0 | 0.3 |
| GPT 4o | 73.1 | 16.1 | 4.9 | 5.9 | 15.3 | 2.2 | 9.4 |
| GPT 4.1 | 60.7 | 27.4 | 1.7 | 10.1 | 20.5 | 0.7 | 18.0 |
| o4-mini | 30.4 | 38.5 | 5.4 | 25.7 | 36.8 | 5.2 | 27.7 |
| o3 | 34.6 | 43.0 | 3.0 | 19.5 | 39.5 | 3.5 | 22.5 |
| Claude Sonnet 3.7 | 41.0 | 35.3 | 4.2 | 19.5 | 27.7 | 2.2 | 29.1 |
| Claude Opus 4.0 | 37.5 | 35.8 | 3.2 | 23.5 | 29.6 | 3.0 | 29.9 |
| Gemini 2.5 Flash | 62.5 | 25.4 | 2.0 | 10.1 | 26.4 | 1.2 | 9.9 |
| DeepSeek V3 | 72.8 | 18.0 | 4.0 | 5.2 | 16.1 | 5.2 | 5.9 |
| Qwen3 235B-A22B-FP8 | 96.3 | 3.5 | 0.3 | 0.0 | 3.7 | 0.0 | 0.0 |

Table 17: Proof failure category distribution across models.

| Category | Claude Sonnet-3.7 | Gemini 2.5 Flash | GPT 4.1 | o4-mini |
|---|---|---|---|---|
| Incomplete Proof (unsolved goals) | 0.55% | 77.55% | 38.24% | 13.10% |
| Cheat Code Usage | 48.53% | 1.20% | 20.55% | 28.30% |
| Unknown Identifier | 15.38% | 5.48% | 9.01% | 12.32% |
| Unknown Tactic | 3.05% | 3.50% | 6.04% | 17.76% |
| Tactic Failed | 9.92% | 3.72% | 2.75% | 6.55% |
| Unknown Constant | 4.14% | 3.94% | 5.05% | 5.55% |
| Syntax Error | 3.16% | 0.77% | 9.56% | 3.66% |
| Type Mismatch | 2.62% | 0.22% | 0.99% | 1.66% |
| Unknown Import | 0.98% | 0.22% | 0.22% | 1.89% |
| Other | 11.67% | 3.40% | 7.58% | 9.21% |

Table 18: Proof failure category distribution on VERINA-A across models.

| Category | Claude Sonnet-3.7 | Gemini 2.5 Flash | GPT 4.1 | o4-mini |
|---|---|---|---|---|
| Incomplete Proof (unsolved goals) | 0.98% | 70.67% | 33.47% | 13.91% |
| Cheat Code Usage | 30.47% | 0.39% | 16.24% | 18.55% |
| Unknown Identifier | 18.55% | 4.13% | 8.51% | 12.30% |
| Unknown Tactic | 4.30% | 6.10% | 7.13% | 20.77% |
| Tactic Failed | 14.45% | 6.10% | 4.55% | 8.27% |
| Unknown Constant | 6.84% | 5.91% | 7.92% | 7.46% |
| Syntax Error | 4.69% | 1.18% | 9.11% | 3.23% |
| Type Mismatch | 4.30% | 0.20% | 1.39% | 1.61% |
| Unknown Import | 1.76% | 0.20% | 0.40% | 2.82% |
| Other | 13.67% | 5.12% | 11.29% | 11.09% |

Table 19: Proof failure category distribution on VERINA-B across models.

| Category | Claude Sonnet-3.7 | Gemini 2.5 Flash | GPT 4.1 | o4-mini |
|---|---|---|---|---|
| Incomplete Proof (unsolved goals) | 0.00% | 86.17% | 44.20% | 12.10% |
| Cheat Code Usage | 71.36% | 2.22% | 25.93% | 40.25% |
| Unknown Identifier | 11.36% | 7.16% | 9.63% | 12.35% |
| Unknown Tactic | 1.48% | 0.25% | 4.69% | 14.07% |
| Tactic Failed | 4.20% | 0.74% | 0.49% | 4.44% |
| Unknown Constant | 0.74% | 1.48% | 1.48% | 3.21% |
| Syntax Error | 1.23% | 0.25% | 10.12% | 4.20% |
| Type Mismatch | 0.49% | 0.25% | 0.49% | 1.73% |
| Unknown Import | 0.00% | 0.25% | 0.00% | 0.74% |
| Other | 9.14% | 1.23% | 2.96% | 6.91% |

# E    CASE STUDIES OF MODEL FAILURES AND EVALUATION METRICS

In this appendix section, we provide a detailed qualitative analysis of common model failure patterns across the three foundational tasks and illustrate how LLMs struggle with different aspects of verifiable code generation through concrete examples. We also discuss how our evaluation metrics flag these failures, highlighting both their effectiveness and limitations.

**Code generation failure: hallucinated method usage.** Figure 18 demonstrates a common LLM failure mode where o4-mini generates code that appears syntactically correct but contains non-existent methods. While the model correctly identifies the XOR-based algorithmic approach and provides accurate comments, it hallucinates the `Int.xor` method that does not exist in Lean 4's standard library. This shows that current LLMs fall short in understanding Lean 4's language features.

```
1    def FindSingleNumber (nums : List Int) (h_precond : FindSingleNumber_precond (nums)) : Int :=
2      -- !benchmark @start code
3      -- XOR all elements: duplicates cancel out, leaving the unique number
4      nums.foldl (fun acc x => Int.xor acc x) 0
5      -- !benchmark @end code
```

Figure 18: Example (`verina_b_1`): o4-mini generates code with hallucinated `Int.xor` method that does not exist in Lean 4.

**Code generation failures: unit test rejections.** Figure 19 illustrates how subtle logical errors in LLM-generated code can lead to unit test failures. The task requires implementing a function that finds the next greater element for each number in `nums1` within the array `nums2`, or outputs -1 if there is none. o4-mini generates a `nextGreaterOne` helper function with a bug in the state management logic. After finding the target element, the function incorrectly calls `aux tl false` instead of `aux tl true` in Line 22, causing it to lose track of having found the target and fail to identify subsequent greater elements. This results in incorrect outputs for the test case where `nums1 = [1, 2, 3]` and `nums2 = [3, 2, 1, 4]` should return `[4, 4, 4]`.

```
1
2    -- !benchmark @start code_aux
3    /-
4    Helper function to find the next greater element of `target` in `nums2`.
5    We traverse the list, first locating `target`, then searching for the first element > `target`.
6    If not found, returns -1.
7    -/
8    def nextGreaterOne (nums2 : List Int) (target : Int) : Int :=
9      let rec aux (l : List Int) (foundTarget : Bool) : Int :=
10       match l with
11       | []        => -1
12       | hd::tl  =>
13           if !foundTarget then
14             if hd == target then
15               aux tl true
16             else
17               aux tl false
18           else
19             if hd > target then
20               hd
21             else
22               aux tl false
23     aux nums2 false
24   -- !benchmark @end code_aux
25
26   def nextGreaterElement (nums1 : List Int) (nums2 : List Int)
27       (h_precond : nextGreaterElement_precond (nums1) (nums2)) : List Int :=
28     -- !benchmark @start code
29     -- For each element in `nums1`, compute its next greater in `nums2`
30     nums1.map fun x => nextGreaterOne nums2 x
31     -- !benchmark @end code
```

Figure 19: Example (`verina_b_57`): o4-mini generates code with logical error in state management, causing unit test failure.

**Specification generation failures: unsound pre-conditions.** Figure 20 demonstrates how LLMs can generate specifications that are too restrictive, leading to unsound pre-conditions. The task description states "Assuming k ≤ number of distinct elements in nums". The ground truth pre-condition correctly uses $k \leq$ `nums.eraseDups.length` to allow k to equal the number of distinct elements. However,

the LLM-generated version uses strict inequality `k < (distinct nums).length`, which incorrectly excludes valid cases where k equals the total number of distinct elements. This makes the pre-condition unsound as it rejects legitimate inputs that should be accepted by the specification. In our test suites, we have a positive test case with `nums = [5]` and `k = 1`. Since the LLM-generated pre-condition rejects this test case, our evaluation metric determines that it is unsound.

```
1   @[reducible, simp]
2   def topKFrequent_precond (nums : List Int) (k : Nat) : Prop :=
3     -- !benchmark @start precond
4     k ≤ nums.eraseDups.length
5     -- !benchmark @end precond
```

(a) Ground truth pre-condition.

```
1   -- !benchmark @start precond_aux
2   @[reducible, simp]
3   def contains (xs : List Int) (x : Int) : Bool :=
4       xs.foldl (fun b y => b || y == x) false
5   @[reducible, simp]
6   def distinct (xs : List Int) : List Int :=
7       xs.foldl (fun acc y => if contains acc y then acc else acc ++ [y]) []
8   -- !benchmark @end precond_aux
9   @[reducible, simp]
10  def topKFrequent_precond (nums : List Int) (k : Nat) : Prop :=
11    -- !benchmark @start precond
12    k < (distinct nums).length
13    -- !benchmark @end precond
```

(b) Unsound pre-condition generated by o4-mini.

Figure 20: Example (`verina_b_76`): o4-mini generates unsound pre-condition using strict inequality instead of allowing k to equal the number of distinct elements.

**Specification generation failures: incomplete pre-conditions.** Figure 21 demonstrates how LLMs can generate overly permissive preconditions that fail to capture essential constraints. The task description specifies that "All integers in both arrays are unique" and that "nums1: A list of integers, which is a subset of nums2". The ground truth precondition correctly enforces three critical requirements: `List.Nodup nums1` ensures uniqueness in the first array, `List.Nodup nums2` ensures uniqueness in the second array, and `nums1.all (fun x => x ∈ nums2)` verifies that `nums1` is indeed a subset of `nums2`. However, the LLM-generated precondition simply uses `True`, completely ignoring all stated constraints. This makes the precondition incomplete as it accepts invalid inputs that violate the problem's fundamental assumptions, potentially leading to incorrect behavior in the implementation and proof generation phases. In our test suites, we have a negative test case with `nums1 = [1, 1]` and `nums2 = [1, 2]`. Since the LLM-generated pre-condition accepts this negative test case, our evaluation metric determines that the LLM-generated pre-condition is incomplete.

```
1   -- Ground truth pre-condition
2   @[reducible, simp]
3   def nextGreaterElement_precond (nums1 : List Int) (nums2 : List Int) : Prop :=
4     -- !benchmark @start precond
5     List.Nodup nums1 ∧
6     List.Nodup nums2 ∧
7     nums1.all (fun x => x ∈ nums2)
8     -- !benchmark @end precond
```

(a) Ground truth pre-condition.

```
1   @[reducible, simp]
2   def nextGreaterElement_precond (nums1 : List Int) (nums2 : List Int) : Prop :=
3     -- !benchmark @start precond
4     True
5     -- !benchmark @end precond
```

(b) Incomplete pre-condition generated by o4-mini.

Figure 21: Example (`verina_advanced_57`): o4-mini generates incomplete pre-condition using `True` instead of enforcing uniqueness and subset constraints.

**Specification generation failures: unsound post-conditions.** Figure 22 illustrates how LLMs can generate post-conditions that miss critical constraints, leading to unsound specifications. The

task involves adding two numbers represented as digit lists in reverse order. The ground truth post-condition correctly enforces three essential properties: arithmetic correctness, digit validity (all digits should be less than 10), and prohibition of leading zeros except for the special case where the result is zero. However, the LLM-generated post-condition omits the leading zero constraint entirely, only checking that the result is non-empty and digits are valid. This unsound specification would accept invalid outputs like `[2, 1, 0]` representing `012` (which is one of the negative test case), which should be rejected in favor of `[2, 1]`. The missing constraint demonstrates how LLMs may capture the primary functional requirements while overlooking more subtle constraints that are crucial for correctness.

```
1    -- !benchmark @start postcond_aux
2    def listToNat : List Nat → Nat
3    | []      => 0
4    | d :: ds => d + 10 * listToNat ds
5    -- !benchmark @end postcond_aux
6    @[reducible, simp]
7    def addTwoNumbers_postcond (l1 : List Nat) (l2 : List Nat) (result: List Nat) (h_precond :
        ↪ addTwoNumbers_precond (l1) (l2)) : Prop :=
8      -- !benchmark @start postcond
9      listToNat result = listToNat l1 + listToNat l2 ∧
10     (∀ d ∈ result, d < 10) ∧
11     -- No leading zeros unless the result is zero
12     (result.getLast! ≠ 0 ∨ (l1 = [0] ∧ l2 = [0] ∧ result = [0]))
13     -- !benchmark @end postcond
```

(a) Ground truth post-condition.

```
1    -- LLM generated post-condition
2    -- !benchmark @start postcond_aux
3    @[reducible, simp]
4    def listToNatRev : List Nat → Nat
5    | []      => 0
6    | d :: ds => d + 10 * listToNatRev ds
7    -- !benchmark @end postcond_aux
8    @[reducible, simp]
9    def addTwoNumbers_postcond (l1 : List Nat) (l2 : List Nat) (result: List Nat) (h_precond :
        ↪ addTwoNumbers_precond (l1) (l2)) : Prop :=
10     -- !benchmark @start postcond
11     result ≠ [] ∧
12     listToNatRev result = listToNatRev l1 + listToNatRev l2 ∧
13     ∀ d, d ∈ result → d < 10
14     -- !benchmark @end postcond
```

(b) Unsound post-condition generated by o4-mini.

Figure 22: Example (`verina_b_5`): o4-mini generates unsound postcondition that fails to rule out leading zeros in the result.

**Specification generation failures: unsound and incomplete post-conditions.** Figure 23 demonstrates how LLMs can generate post-conditions that are both unsound and incomplete by failing to handle edge cases properly. The task involves finding the smallest single-digit prime factor of a natural number. The ground truth post-condition correctly handles all cases including the edge case where `n = 0`, specifying that the result should be 0 when the input is 0 or when no single-digit prime divides n. However, the LLM-generated post-condition fails to consider `n = 0` entirely. When `n = 0`, the condition `n % p ≠ 0` is false for any prime p (since `0 % p = 0`), making the first disjunct impossible to satisfy. This renders the specification both unsound (accepts incorrect outputs) and incomplete (rejects valid cases where `n = 0`). The missing edge case handling demonstrates how LLMs may overlook corner cases that are crucial for specification completeness (and soundness). We have a positive test case where `n = 0` and `result = 0` and a corresponding negative test case where `n = 0` and `result = 2` that capture this edge case. The LLM-generated post-condition rejects the positive test case and accepts the negative test case, therefore our evaluation metric determines that this generated post-condition is both unsound and incomplete.

```
1   -- Ground truth post-condition
2   @[reducible, simp]
3   def singleDigitPrimeFactor_postcond (n : Nat) (result: Nat) (h_precond : singleDigitPrimeFactor_precond (
        ↪ n)) : Prop :=
4     -- !benchmark @start postcond
5     result ∈ [0, 2, 3, 5, 7] ∧
6     (result = 0 → (n = 0 ∨ [2, 3, 5, 7].all (n % · ≠ 0))) ∧
7     (result ≠ 0 → n ≠ 0 ∧ n % result == 0 ∧ (List.range result).all (fun x => x ∈ [2, 3, 5, 7] → n % x
        ↪ ≠ 0))
8     -- !benchmark @end postcond
```

(a) Ground truth post-condition.

```
1   -- LLM generated post-condition
2   -- !benchmark @start postcond_aux
3   @[reducible, simp]
4   def isSingleDigitPrime (p : Nat) : Prop :=
5       p = 2 ∨ p = 3 ∨ p = 5 ∨ p = 7
6   -- !benchmark @end postcond_aux
7   @[reducible, simp]
8   def singleDigitPrimeFactor_postcond (n : Nat) (result: Nat) (h_precond : singleDigitPrimeFactor_precond (
        ↪ n)) : Prop :=
9     -- !benchmark @start postcond
10    -- Either no small prime divides n, so we return 0
11    (result = 0 ∧ ∀ p, isSingleDigitPrime p → n % p ≠ 0)
12    ∨
13    -- Or result is the smallest -singledigit prime divisor of n
14    (isSingleDigitPrime result ∧ n % result = 0 ∧ ∀ q, isSingleDigitPrime q → n % q = 0 → result ≤ q)
15    -- !benchmark @end postcond
```

(b) Unsound and incomplete post-condition generated by o4-mini.

Figure 23: Example (`verina_b_72`): o4-mini generates unsound and incomplete post-condition that fails to handle the edge case `n = 0`.

**Untestable post-conditions.** Figure 24 demonstrates the limitations of our testing-based evaluation framework when encountering specifications with quantifiers over complicated structures or infinite domains. The LLM-generated post-condition for finding the length of the longest increasing subsequence contains a universal quantifier $\forall$ `s : List Int` that ranges over all possible integer lists, making it impossible to evaluate even with plausible testing. Our evaluation framework returns unknown for such cases, as neither decidable testing nor plausible exploration can adequately handle the unbounded quantification. This example highlights a fundamental challenge in automatically evaluating LLM-generated formal specifications: while our framework successfully handles most practical cases, very complicated specifications require more comprehensive approaches such as automated theorem provers or LLM-based proof generation, which we leave to future work.

```
1   -- !benchmark @start postcond_aux
2   @[reducible, simp]
3   def IsSubsequence : List Int → List Int → Prop
4   | [], _      => True
5   | _ :: _, [] => False
6   | x :: xs, y :: ys =>
7       if x = y then IsSubsequence xs ys
8       else IsSubsequence (x :: xs) ys
9
10  @[reducible, simp]
11  def strictlyIncreasing : List Int → Prop
12  | []          => True
13  | [_]         => True
14  | x :: y :: rest => x < y ∧ strictlyIncreasing (y :: rest)
15  -- !benchmark @end postcond_aux
16  @[reducible, simp]
17  def lengthOfLIS_postcond (nums : List Int) (result: Nat) (h_precond : lengthOfLIS_precond (nums)) : Prop :=
18    -- !benchmark @start postcond
19    (∀ s : List Int, IsSubsequence s nums ∧ strictlyIncreasing s → List.length s ≤ result)
20    ∧ ∃ s : List Int, IsSubsequence s nums ∧ strictlyIncreasing s ∧ List.length s = result
21    -- !benchmark @end postcond
```

Figure 24: Example (`verina_b_25`): o4-mini generates post-condition with quantifiers over lists that cannot be evaluated by plausible testing.

**Proof generation success with iterative refinement.** Figure 25, 26, 27, and 28 demonstrate o4-mini's iterative proof refinement process over 24 attempts. The task involves proving that a list element removal function satisfies its specification. Initially, o4-mini generates an overly simple proof using direct induction and simp tactics, which fails to handle the complex logical structure. After 23 failed refinement attempts, the model makes useful improvements based on Lean compiler error

messages, significantly restructuring its approach. The error message from the 23rd attempt reveals that the proof got stuck on conditional expressions involving decide predicates. In the successful 24th attempt, o4-mini improves its strategy by introducing an auxiliary lemma `helper_eq_filter` and using explicit case analysis with `by_cases`. This demonstrates how iterative refinement with compiler feedback can guide LLMs toward substantially different and correct proof strategies, though it may require many attempts to achieve success.

```
1   @[reducible, simp]
2   def removeElement_precond (lst : List Nat) (target : Nat) : Prop :=
3     -- !benchmark @start precond
4     True
5     -- !benchmark @end precond
6
7   def removeElement (lst : List Nat) (target : Nat) (h_precond : removeElement_precond (lst) (target)) :
        ↪ List Nat :=
8     -- !benchmark @start code
9     let rec helper (lst : List Nat) (target : Nat) : List Nat :=
10      match lst with
11      | [] => []
12      | x :: xs =>
13        let rest := helper xs target
14        if x = target then rest else x :: rest
15    helper lst target
16    -- !benchmark @end code
17
18  @[reducible]
19  def removeElement_postcond (lst : List Nat) (target : Nat) (result: List Nat) (h_precond :
        ↪ removeElement_precond (lst) (target)): Prop :=
20    -- !benchmark @start postcond
21    result = lst.filter (fun x => x ≠ target)
22    -- !benchmark @end postcond
23
24
25  -- !benchmark @start proof_aux
26  -- !benchmark @end proof_aux
27
28  theorem removeElement_spec_satisfied (lst: List Nat) (target: Nat) (h_precond : removeElement_precond (
        ↪ lst) (target)):
29      removeElement_postcond (lst) (target) (removeElement (lst) (target) h_precond) h_precond := by
30    -- !benchmark @start proof
31    -- Unfold the definitions of removeElement and the postcondition
32    dsimp [removeElement, removeElement_postcond]
33    -- Proceed by induction on the input list
34    induction lst with
35    | nil =>
36    -- Base case: both sides reduce to []
37    simp
38    | cons x xs ih =>
39    -- Inductive step: use the induction hypothesis
40    simp [ih]
41    -- !benchmark @end proof
```

Figure 25: Example (`verina_b_64`): o4-mini generates incorrect proof on the first attempt.

```
1   -- !benchmark @start proof_aux
2   -- Auxiliary lemma: the `helper` function produces exactly `lst.filter (fun x => x ≠ target)`
3   theorem helper_eq_filter (lst : List Nat) (target : Nat) :
4       removeElement.helper lst target = lst.filter (fun x => x ≠ target) := by
5     induction lst with
6     | nil =>
7       -- both sides reduce to `[]`
8       simp [removeElement.helper]
9     | cons x xs ih =>
10      -- unfold one step of `helper`
11      dsimp [removeElement.helper]
12      -- solve the `if` statement using induction hypothesis
13      simp [ih]
14  -- !benchmark @end proof_aux
15
16  theorem removeElement_spec_satisfied (lst: List Nat) (target: Nat) (h_precond : removeElement_precond (
        ↪ lst) (target)):
17      removeElement_postcond (lst) (target) (removeElement (lst) (target) h_precond) h_precond := by
18    -- !benchmark @start proof
19    -- unfold the postcondition and the definition of `removeElement`
20    dsimp [removeElement_postcond, removeElement]
21    -- the precondition is trivial
22    clear h_precond
23    -- apply the auxiliary lemma to conclude
24    exact helper_eq_filter lst target
25    -- !benchmark @end proof
```

Figure 26: Example (`verina_b_64`): o4-mini generates incorrect proof after 23 refinement attempts.

```
1   unsolved goals
2   case cons
3   target x : Nat
4   xs : List Nat
5   ih : removeElement.helper xs target = List.filter (fun x => decide (x ≠ target)) xs
6   ⊢ (if x = target then List.filter (fun x => !decide (x = target)) xs
7     else x :: List.filter (fun x => !decide (x = target)) xs) =
8     List.filter (fun x => !decide (x = target)) (x :: xs)
```

Figure 27: Example (`verina_b_64`): Lean 4 compiler error message from the 23rd refinement attempt.

```
1   -- !benchmark @start proof_aux
2   -- Auxiliary lemma: the `helper` function produces exactly `lst.filter (fun x => x ≠ target)`
3   theorem helper_eq_filter (lst : List Nat) (target : Nat) :
4       removeElement.helper lst target = lst.filter (fun x => x ≠ target) := by
5     induction lst with
6     | nil =>
7       -- both sides reduce to `[]`
8       simp [removeElement.helper]
9     | cons x xs ih =>
10      -- unfold one step of `helper`
11      dsimp [removeElement.helper]
12      -- case-split on whether `x = target`
13      by_cases h : x = target
14      case pos =>
15        -- if `x = target`, `helper` drops it, and `filter` also drops it
16        simp [h, ih]
17      case neg =>
18        -- if `x ≠ target`, `helper` keeps it, and `filter` also keeps it
19        simp [h, ih]
20  -- !benchmark @end proof_aux
21
22  theorem removeElement_spec_satisfied (lst: List Nat) (target: Nat) (h_precond : removeElement_precond (
        ↪ lst) (target)):
23      removeElement_postcond (lst) (target) (removeElement (lst) (target) h_precond) h_precond := by
24    -- !benchmark @start proof
25    -- unfold the postcondition and the definition of `removeElement`
26    dsimp [removeElement_postcond, removeElement]
27    -- the precondition is trivial
28    clear h_precond
29    -- apply the auxiliary lemma to conclude
30    exact helper_eq_filter lst target
31    -- !benchmark @end proof
```

Figure 28: Example (`verina_b_64`): o4-mini generates correct proof on the 24th attempt.

# F  USE OF LLM

LLMs were used in a limited and supervised manner during the construction of VERINA. Specifically, we employed OpenAI o3-mini with few-shot prompting to assist in translating 59 Dafny instances from CloverBench (Sun et al., 2024) into Lean, as discussed in Section 3.2. All such translations were subsequently inspected, corrected, and verified by the authors to ensure accuracy. In addition, LLMs were used as assistive tools for editing and polishing the presentation of the paper. LLMs were not involved in research ideation, discovery of related work, experimental design, dataset selection, or analysis.

