# OpenReview forum: "VERINA: Benchmarking Verifiable Code Generation"
_ICLR.cc/2026/Conference — ICLR 2026 Poster_

### Official Review · Reviewer_a4Ez · 2025-10-30

**Soundness:** 2
**Presentation:** 3
**Contribution:** 2
**Rating:** 4
**Confidence:** 4

**Summary:**

This paper introduces VERINA (Verifiable Code Generation Arena), a benchmark comprising 189 manually curated programming tasks in Lean for evaluating end-to-end verifiable code generation. The benchmark assesses three foundational tasks—code generation (CodeGen), specification generation (SpecGen), and proof generation (ProofGen)—along with their flexible combinations. The best model (o4-mini) achieves 61.4% code correctness, 51.0% specification soundness/completeness, and only 3.6% proof success.

**Strengths:**

- Clear Presentation: The presentation is well-executed.
- Extensive Task Support: VERINA comprehensively supports all three core tasks: CodeGen, SpecGen, and ProofGen, utilizing a modular and compositional approach.
- Thorough Experimental Evaluation: The evaluation is extensive, involving 8 general-purpose and 3 specialized LLMs. It includes detailed ablations on the impact of contextual information, iterative refinement, and a breakdown of difficulty levels.

**Weaknesses:**

- Limited Language/Tool Coverage: The reliance on Lean significantly restricts the generalizability of the findings. The paper fails to justify how Lean-specific results would translate to other widely used verification ecosystems like Dafny, Verus, or Frama-C, which often feature different proof automation capabilities and specification styles.
- Unclear Justification for Lean in Code Verification: Lean is primarily known for mathematical verification rather than production code verification. The paper does not adequately explain its choice of Lean for this context.
- Insufficient Analysis of Proof Generation Failures: Despite proof generation being a major hurdle (with a mere 3.6% success rate, the lowest among all tasks), the paper offers minimal systematic analysis of the underlying causes of failure or actionable insights.
- Task Simplicity: The tasks appear to be overly simplistic. The paper should provide information on the average number of branches per task.

**Questions:**

- What evidence demonstrates Lean is used for production code verification (not mathematics)? How do you justify that insights from ITP-based verification in Lean generalize to ATP-based systems (Dafny, Verus, Frama-C) that dominate production use?
- Why not include at least one ATP-based system for comparison to validate generalizability?
- Can you provide systematic breakdown of proof failure types (wrong strategy, missing lemmas, syntax errors, etc.) with percentages?
- How do LLM-generated proof structures differ from the 46 human-written ground truth proofs? Are the remaining tasks really provable in Lean?
- What is the actual end-to-end success rate (NL → verified code) for each model?
- What are the average branches of the task?



Comments:
- Incorrect citations: some paper references are incorrect, like SAFE, AutoVerus, etc.

---

> ### Author Response · Authors · 2025-11-21
>
> We thank the reviewer for their insightful and constructive review and address their questions and feedback below.We will post a revised manuscript on OpenReview based on the feedback and additional experiments within the next few days. (`Qi` stands for question `i`)
>
> **Q1: What evidence demonstrates Lean is used for production code verification?**
>
> Lean is actively used for production code verification beyond mathematics:
> - AWS production systems [1]: AWS uses Lean extensively as verification infrastructure for critical projects including Cedar (cloud services authorization verification), LNSym (symbolic simulator for Armv8 machine-code programs), and SampCert (a verified implementation of randomized algorithms);
> - Rust verification: Aeneas [2] verifies Rust programs by translating them into Lean for formal verification; Cedar [3] models Rust programs in Lean with differential testing;
> - Blockchain smart contract verification: Clear [4] enables proving properties about Ethereum smart contracts using Lean;
> - Computer science formalization: CSLib [5] represents a joint effort across universities (Stanford, UT Austin) and major companies (Google, AWS) to formalize undergraduate-level computer science in Lean, providing foundations for program verification beyond mathematics.
>
> These projects demonstrate Lean's growing adoption for practical software verification in production environments. We will revise the manuscript to include these examples and better justify Lean's suitability for code verification.
>
> [1] Leo de Moura, How the Lean language brings math to coding and coding to math. Amazon Science Blog. 2024
>
> [2] Ho et al., Aeneas: Rust Verification by Functional Translation. arXiv:2206.07185.
>
> [3] Mike Hikes, How we built Cedar with automated reasoning and differential testing. Amazon Science Blog. 2023.
>
> [4] NethermindEth, Clear: Interactive formal verification tool for Yul programs. 2025.
>
> [5] Barrett et al., CSLib: A Foundation for Computer Science in Lean 4. 2025.
>
> **Q2: How do insights from ITP-based verification in Lean generalize to ATP-based systems?**
>
> While Lean’s syntax differs from other ecosystems such as Dafny and Verus, the core challenges in verifiable code generation are shared. Both ITP and ATP frameworks require: (i) generating correct code and sound/complete specifications, and (ii) identifying key properties such as loop invariants for constructing proofs. Therefore, insights from VERINA on these core capabilities are therefore broadly relevant to ATP-based systems.
>
> We agree that including ATP-based systems like Dafny or Verus would strengthen VERINA’s generalizability, but this requires significant effort. Therefore, we will pursue this as important future work. We will revise the manuscript to discuss the differences and commonalities of these verification ecosystems, improving our explanation on how VERINA's findings provide broader insights across both ITP and ATP ecosystems.
>
> **Q3: Can you provide more systematic analysis of proof generation failures?**
>
> We acknowledge that while Appendix C provides case studies of proof generation failures, the paper would benefit from more systematic analysis in the main text. Following this feedback, we have conducted comprehensive categorization of all proof failures across models:
>
> | Category | Claude Sonnet-3.7 | Gemini 2.5 Flash | GPT 4.1 | o4-mini |
> |----------|----------:|----------:|----------:|----------:|
> | Incomplete Proof (unsolved goals) | 0.55% | 77.55% | 38.24% | 13.10% |
> | Cheat Code Usage | 48.53% | 1.20% | 20.55% | 28.30% |
> | Unknown Identifier | 15.38% | 5.48% | 9.01% | 12.32% |
> | Unknown Tactic | 3.05% | 3.50% | 6.04% | 17.76% |
> | Tactic Failed | 9.92% | 3.72% | 2.75% | 6.55% |
> | Unknown Constant | 4.14% | 3.94% | 5.05% | 5.55% |
> | Syntax Error | 3.16% | 0.77% | 9.56% | 3.66% |
> | Type Mismatch | 2.62% | 0.22% | 0.99% | 1.66% |
> | Unknown Import | 0.98% | 0.22% | 0.22% | 1.89% |
> | Other | 11.67% | 3.40% | 7.58% | 9.21% |
>
> The analysis reveals distinct failure patterns across models: incomplete proofs with unsolved goals dominate for Gemini (77.55%) and GPT 4.1 (38.24%), indicating these models struggle to develop complete proof strategies beyond initial tactics, while Claude (48.53%) and o4-mini (28.30%) frequently resort to cheat codes (sorry, admit, axiom) suggesting they recognize inability to complete proofs. Unknown identifiers, tactics, and constants collectively account for 15-30% of failures across models, demonstrating lack of familiarity with Lean's standard library and tactic ecosystem. We will add full systematic analysis to the revised manuscript, providing researchers with clear direction for improving LLM-based theorem provers under formal verification scenarios.

---

> ### Author Response · Authors · 2025-11-21
>
> **Q4.1: How do LLM-generated proof structures differ from the 46 human-written ground truth proofs?**
>
> Regarding the proof structure difference, we analyzed successful LLM-generated proofs and found significant verbosity increases with iterative refinement. After manual inspection, we observed that LLM-generated proofs tend to be more verbose, while human proofs leverage automation tactics (e.g., `simp`, `aesop`) more effectively. Moreover, LLMs often cannot correctly identify or use relevant lemmas from the standard library, leading them to explicitly prove tedious intermediate goals that humans would automate. This suggests improving LLMs' understanding of proof automation and lemma usage is a critical direction for future work. We will revise the manuscript to discuss this comparison.
>
> | Refinement Iterations | Average Proof Length |
> |----------------------|------------------:|
> | Human (ground truth) | 169.6 |
> | ≤1 (direct generation) | 171.9 |
> | ≤8 | 217.7 |
> | ≤16 | 283.1 |
> | ≤32 | 367.7 |
> | ≤64 | 431.4 |
>
> **Q4.2: Are the remaining tasks really provable in Lean?**
>
> Regarding the provability of the remaining tasks, we believe all tasks in VERINA are provable in Lean. As detailed in Section 3.2, we manually validated all code implementations and specifications to ensure correctness, with comprehensive test suites achieving 100% code coverage and all ground truth specifications passing their respective test cases. Theoretically, Lean's expressiveness with higher-order logic guarantees it is suitable for formal verification of these tasks. Empirically, our evaluation demonstrates that LLMs already successfully generate proofs for tasks beyond the 46 samples with ground truth proofs, confirming these problems are provable. Moreover, benchmarks like PutnamBench [1] show that LLMs are rapidly catching up on challenging mathematical proving tasks, suggesting similar progress might happen for VERINA.
>
> [1] Tsoukalas et al., PutnamBench: Evaluating Neural Theorem-Provers on the Putnam Mathematical Competition. 2024.
>
> **Q5: Can you report results for the full end-to-end task?**
>
> Yes. Below, we show the end-to-end verifiable code generation results where models generate code, specification, and proof. The end-to-end results show similar patterns to individual task performance, where ProofGen tasks remain the primary bottleneck in verifiable code generation. We will add full end-to-end results to the manuscript.
>
> | Model | Code Pass Rate | Spec Pass Rate | Code+Spec Pass Rate | E2E Pass Rate |
> |-------|---------------:|---------------:|--------------------:|--------------------------:|
> | o4-mini | 60.43% | 45.45% | 32.62% | 3.21% |
> | GPT 4.1 | 56.68% | 39.57% | 29.41% | 2.67% |
> | Claude Sonnet 3.7 | 42.25% | 41.18% | 22.99% | 1.60% |
> | Gemini 2.5 Flash | 37.97% | 40.64% | 26.74% | 2.14% |
>
> **Q6: What are the average branches of the tasks?**
>
> The average number of branches per task (in its python ground truth, counting loop as one branch) is 1.5, with 0 minimum and 10 maximum. However, we believe branch count is not a perfect metric for complexity in formal verification contexts. For example, even though a loop counts as only 1 branch, it is significantly challenging to prove since LLMs must identify and prove loop invariants, which requires deep formal reasoning about program behavior. While individual tasks may appear simple, our evaluation shows they pose significant challenges on SpecGen and ProofGen, especially on the VERINA-B subset. This demonstrates that VERINA addresses fundamental challenges that must be solved before tackling more complex scenarios, making it an appropriate and valuable benchmark for current capabilities. We will revise the manuscript to further clarify these design choices and scope considerations.
>
> **Q7: Incorrect citations.**
>
> We thank the reviewer for pointing this out. We will fix all inaccurate citations in the revised manuscript.

---

> ### Author Response · Authors · 2025-11-28
>
> Thank you for the constructive feedback. We have provided detailed responses to all your questions and have conducted the additional experiments you requested. We would greatly appreciate any additional feedback you might have on our responses.

---

> ### Author Response · Authors · 2025-12-03
>
> We have revised our manuscript to address all the mentioned concerns. The changes are highlighted in blue. Specifically, we did the following changes (all line numbers are from the revised version):
> - Q1, Q2: We added Appendix A to discuss the justification of choice of Lean and the scope and generalizability of VERINA;
> - Q3: We added the quantitative breakdown of code/spec/proof failure mode categorized Appendix D. We updated/added line 1562-1570 and Table 8-19.
> - Q4: We added experiment results and discussion in Appendix D line 1498-1507 and Figure 16;
> - Q5: We added the end-to-end result and discussion in Section 5 line 435-452;
> - Q6: We added clarifying sentences in Section 6 527-528;
> - Q7: We corrected the citations.

---

### Official Review · Reviewer_6DqP · 2025-10-31

**Soundness:** 2
**Presentation:** 3
**Contribution:** 2
**Rating:** 4
**Confidence:** 3

**Summary:**

This paper introduces VERINA (Verifiable Code Generation Arena), a new benchmark designed to evaluate the ability of Large Language Models (LLMs) to perform verifiable code generation. The authors define this as the joint task of generating code, formal specifications, and proofs that the code aligns with the specifications. The benchmark consists of 189 manually curated programming tasks in the Lean language, each with detailed descriptions, reference implementations, formal specifications, and comprehensive test suites. The authors evaluated state-of-the-art LLMs on VERINA and found that these models face significant challenges, especially in proof generation. The paper concludes that VERINA provides a rigorous framework for measuring and advancing LLM capabilities in producing formally verified software.

**Strengths:**

1). VERINA covers CodeGen, SpecGen, and ProofGen and allows their compositions, which matches realistic verification workflows rather than isolated tasks.

2). The formal treatment of soundness and completeness for pre- and post-conditions is crisp. The two-stage evaluator (prover first, then property-based testing) gives a practical way to score imperfectly decidable relationships and communicates uncertainty via “R might hold” and error bars.

**Weaknesses:**

1). Modest scale and representativeness are not convincing. With 189 single-file tasks, the benchmark remains small for broad performance claims. The focus on stand-alone snippets under-represents the multi-module, dependency-heavy settings that matter for real systems. The paper acknowledges this limitation but it still constrains the practical impact.

2). The “100% coverage” is established on Python references rather than the Lean artifacts under evaluation, due to missing Lean tooling. This creates a fidelity gap between what is instrumented and what is graded. Mutation analysis or Lean-side coverage proxies would reduce this gap.

3). Sources include MBPP, LeetCode, and LiveCodeBench. The paper notes risk but does not present a systematic decontamination audit or overlap analysis against model training corpora. This could inflate performance on familiar patterns. Besides, these sources from popular platforms are almost certainly present in the training corpora of the state-of-the-art LLMs evaluated in the study.

4). This contamination risk in  3) may directly undermine the evaluation of the "CodeGen" task. The paper reports that the best model, 04-mini, achieves a 61.4% code correctness rate. This relatively high score—especially when contrasted with the dismal 3.6% proof success rate—is ambiguous. It may not reflect the model's genuine ability to generate correct Lean code from a description, but rather its ability to recall a memorized algorithm and "translate" it into the required Lean syntax.

5). The central premise of VERINA is to provide a "holistic evaluation framework" for the joint task of generating code, specifications, and proofs. However, if the CodeGen task is compromised by memorization, the benchmark is no longer evaluating a "generation" pipeline. Instead, it may be evaluating a disjointed task: recalling code, and then attempting to generate specifications and proofs for that recalled code. This is a fundamentally different and less challenging task than true end-to-end generation from a novel problem description, which is what the paper claims to benchmark. Also, the contamination risk makes the paper's main findings difficult to interpret. The key result:  ProofGen is the primary bottleneck, is called into question. Is proof generation difficult in general, or is it difficult when the associated code is recalled from memory? The latter is a much weaker claim and provides less insight. The benchmark fails to isolate whether the models are failing at reasoning or simply failing to formally verify a solution they do not understand but have memorized.

6). The evaluator may return “R might hold” when proofs are inconclusive and tests pass. This is reasonable, yet it weakens the claim of verifiable benchmarking; readers would benefit from a quantitative breakdown of proof-decided vs. test-backed cases and sensitivity to the testing budget.

In general, the proposed SpecGen metric and the compositional framing are valuable. However, the dataset scale is modest, coverage is measured on proxies, “R might hold” weakens formal guarantees without a thorough sensitivity analysis, and contamination controls are limited. The high probability of data contamination is critical. It strikes at the construct validity of the benchmark, making it unclear what is actually being measured. For a paper whose primary contribution is a new evaluation benchmark, this ambiguity regarding what it validly measures renders its conclusions unreliable and its utility to the community questionable. Addressing these issues may lift the work to a acceptance.

**Questions:**

1). How sensitive are “R might hold” outcomes to the property-based testing budget and random seeds? Please report the split of “proved vs. tested vs. unknown,” and provide seed-averaged results.

2). Beyond Python, did you attempt Lean-level mutation testing or adversarial test synthesis to ensure that the Lean implementations are exercised comparably to the Python references?

3). Can you provide n-gram or signature overlap checks between VERINA items and public code corpora likely used for pretraining? A filtered subset with low overlap would help calibrate headline metrics.

4). What is the marginal utility curve of iterative refinement (cost vs. accuracy) and how does Copra’s benefit scale across difficulty strata? A budget-normalized comparison would help practitioners.

5). Were prompts fully identical across models for each task? If not, please quantify the effect of template differences on pass@1 and pass@k.

6). Could you report a control where the code is summarized into behavior-only comments or contracts before SpecGen to test whether verbosity or implementation detail confounds the result?

---

> ### Author Response · Authors · 2025-11-21
>
> We thank the reviewer for their insightful and constructive review and address their questions and feedback below.We will post a revised manuscript on OpenReview based on the feedback and additional experiments within the next few days. (`Wi` stands for weakness `i`; `Qi` stands for question `i`)
>
> **Q1 & W6: How sensitive are "R might hold" outcomes to property-based testing budget and random seeds? Please report the split of "proved vs. tested vs. unknown" and provide seed-averaged results.**
>
> Following this feedback, we have conducted sensitivity analysis on o4-mini SpecGen results by varying the property-based testing budget (the number of generated tests) and averaging across 5 random seeds. The results shown in the following table demonstrate that VERINA's choice of 1000 instances is sufficient, as increasing the budget to 2000 instances provides minimal additional benefit, and the standard deviations across seeds are small. All evaluation components contribute meaningfully to the final determination, demonstrating the comprehensiveness of our specification evaluation approach. The most variable component (property-based testing) contributes less than 13% of cases. As LLM-based theorem provers continue to improve, we expect the "Guaranteed to Hold (proved)" percentage to increase, providing more formal guarantees to the SpecGen evaluation. We will add these sensitivity analyses to the revised manuscript.
>
> | # Test Budget | Guaranteed to Hold (proved) % | Might Hold (unit tests) % | Might Hold (property-based testing) % | Guaranteed to Not Hold (counter example found) % | Unknown % | Cannot Compile % |
> |-------------:|--------:|--------:|--------:|-----------:|------------------:|-----:|
> | 10          | 3.7% | 38.7% | 2.1%       | 21.0%       | 10.5% ± 0.412%          | 24.1% |
> | 100        | 3.7% | 38.7% | 5.6%       | 21.3%       | 7.3% ± 0.594%         | 24.1% |
> | 1000       | 3.7% | 38.7% | 5.7%       | 21.3%       | 7.2% ± 0.134%         | 24.1% |
> | 2000       | 3.7% | 38.7% | 5.7%       | 21.3%       | 7.2% ± 0.114%         | 24.1% |
>
> Where "Guaranteed to Hold (proved)" means the relationship was formally verified using LLM-based theorem provers, "Might Hold (unit tests)" means the specification passes all unit tests without property-based testing, "Might Hold (property-based testing)" means no counterexample was found through additional property-based testing with plausible, "Guaranteed to Not Hold" means a counterexample was found (either test case itself or the property-based testing), "Unknown" means the evaluator could not determine the outcome, and "Cannot Compile" means the generated specification has syntax errors.
>
> **Q2 & W2: Does your test suite achieve 100% coverage in Lean reference implementations?**
>
> Yes. We acknowledge the reviewer’s concern that achieving 100% coverage in Python does not theoretically guarantee the same in Lean. To validate this in practice, we manually inspected all benchmark instances and executed our test suites on the Lean ground-truth implementations. We confirmed that our test suites also achieve 100% coverage in our Lean ground truth code. This is feasible because, for the problems we consider, cross-language test suites are typically sufficient to assess functional correctness across language implementations, which is a common practice adopted by LeetCode and prior work on evaluating LLM-generated code [1, 2]. Our additional manual inspection further strengthens the benchmark quality. We will revise the paper to clarify this.
>
> [1] Cassano et al. MultiPL-E: A Scalable and Extensible Approach to Benchmarking Neural Code Generation. IEEE TSE, 2023.
>
> [2] Roziere et al. Leveraging Automated Unit Tests for Unsupervised Code Translation. ICLR, 2022.

---

> ### Author Response · Authors · 2025-11-21
>
> **Q3 & W3~5: Are data contamination risks properly mitigated in VERINA?**
>
> We thank the reviewers for pointing out this very important concern. We believe contamination is not a significant issue in VERINA, and we answer the reviewer's detailed concerns point by point.
>
> *1. N-gram overlap analysis of direct contamination*
>
> One of the potential and critical contamination is that VERINA’s Lean solutions appear in training corpora, which could undermine the evaluation. We thank the reviewer for pointing out this important problem and address this direct contamination concern by conducting N-gram overlap analysis between our Lean ground truth and the full bigcode/the-stack pretraining dataset [1], which contains ~550M rows of coding files obtained from GitHub. The results show no overlap on 10-gram comparison (following standard decontamination practices like Qwen 2.5 Coder [2]), confirming that VERINA's Lean artifacts are novel and not present in public pretraining corpora. This analysis reveals that VERINA’s CodeGen has minimal risk of data contamination, and VERINA's evaluation across all tasks (CodeGen, SpecGen, ProofGen) reflects genuine model capabilities.
>
> *2. Evaluation results reveal fundamental limitations in verifiable code generation, despite potential algorithm familiarity*
>
> The reviewer has concerns that "contamination...may directly undermine the evaluation" and that the benchmark would evaluate "recalling code" rather than "true end-to-end generation". In verifiable code generation LLM are required to perform specification and proof generation beyond code generation, which are fundamentally different skills. Our evaluation shows that even for algorithmically familiar problems, the best LLMs struggle significantly with formal specification generation (51% success rate for best model) and proof generation (3.6% success rate), demonstrating that memorized algorithmic solutions do not transfer to verification tasks in current LLMs. The stark contrast between 61.4% CodeGen and 3.6% ProofGen success rates, despite both potentially benefiting from algorithmic familiarity, demonstrates that LLMs cannot formally reason about algorithms they may have seen and thus suggest a lack of deep understanding of the algorithm they use.
>
> *3. ProofGen evaluation explicitly isolates formal reasoning from memorization*
>
> Our benchmark evaluation design directly addresses the reviewer’s concern on “Is proof generation difficult in general, or is it difficult when the associated code is recalled from memory?”. As shown in Figure 2 and detailed in Section 4.1, our ProofGen task provides ground truth code and specifications to the model, explicitly isolating proof construction capability from any code generation or memorization factors. The 3.6% success rate (OpenAI o4-mini) directly measures formal reasoning ability, not the ability to verify memorized solutions. This design ensures that the finding "ProofGen is the primary bottleneck" reflects genuine limitations in formal reasoning, providing meaningful insights regardless of algorithmic familiarity.
>
> We will revise the manuscript to better explain how verification tasks evaluated in VERINA differ from simple code generation and why contamination concerns are mitigated, including the n-gram overlap analysis.
>
> [1] Kocetkov et al., The Stack: 3 TB of permissively licensed source code. Arxiv, 2022.
>
> [2] Hui et al., Qwen2.5-Coder Technical Report. Arxiv, 2024.

---

> ### Author Response · Authors · 2025-11-21
>
> **Q4: What is the marginal utility curve of iterative refinement (cost vs. accuracy) and how does Copra's benefit scale across difficulty strata? Can you provide a budget-normalized comparison?**
>
> Following this feedback, we have conducted cost-effectiveness analysis of iterative refinement on OpenAI o4-mini stratified by different dataset origin (VERINA-A vs VERINA-B, as defined in Appendix B):
> | Token Budget | Overall Proof Success | VERINA-A Proof Success | VERINA-B Proof Success |
> |--------------:|----------------------:|------------------------:|------------------------:|
> | 50,000 | 8.99% | 13.89% | 2.47% |
> | 100,000 | 11.11% | 16.67% | 3.70% |
> | 150,000 | 12.70% | 19.44% | 3.70% |
> | 200,000 | 13.23% | 20.37% | 3.70% |
> | 250,000 | 13.76% | 21.30% | 3.70% |
> | 300,000 | 14.29% | 22.22% | 3.70% |
> | 350,000 | 14.29% | 22.22% | 3.70% |
>
> Average successful proof costs 57,594 tokens (maximum 288,563 tokens per task for successful proofs; maximum 730,600 tokens per task across all attempts up to 64 refinement times including failures).
>
> In comparison, here is the cost-effectiveness analysis of COPRA using OpenAI o4-mini as the base model:
> | Token Budget | Overall Proof Success | VERINA-A Proof Success | VERINA-B Proof Success |
> |--------------:|---------:|----------:|----------:|
> | 50,000 | 4.76% | 8.33% | 0.00% |
> | 100,000 | 5.82% | 9.26% | 1.23% |
> | 150,000 | 6.35% | 10.19% | 1.23% |
> | 200,000 | 6.88% | 11.11% | 1.23% |
> | 250,000 | 6.88% | 11.11% | 1.23% |
> | 300,000 | 7.41% | 12.04% | 1.23% |
> | 350,000 | 7.94% | 12.96% | 1.23% |
>
> Average successful proof costs 84,027 tokens (maximum 322,784 tokens per task for successful proofs; maximum 569,176 tokens per task across all attempts including failures).
>
> We will add these analyses to the manuscript with detailed cost-benefit curves and performance breakdowns.
>
> **Q5: Were prompts fully identical across models for each task?**
>
> As detailed in Appendix A.3, we employed identical 2-shot DSPy structural prompting across all general-purpose models. However, for DeepSeek Prover V2 7B and Goedel Prover V2 32B on ProofGen, we used their specialized prompt templates as these models were trained only on these templates. To quantify the impact, we tested these provers with standard DSPy prompts and found they achieved 0% success on ProofGen due to instruction-following issues (inability to format outputs for parsing). This justifies our use of specialized prompts to ensure fair evaluation of their actual proving capabilities. We will further clarify in Section 5 that specialized provers used their recommended prompts while all general-purpose models were evaluated with identical prompts for direct comparability.
>
> **Q6: Could you test whether code verbosity or implementation details confound SpecGen results by using behavior-only summaries instead of full code?**
>
> To address this concern, we conducted an ablation study where we replaced the full code reference with behavior-only summaries (generated by prompting LLMs to extract high-level contracts and behavior descriptions without implementation details). The results show highly model-dependent effects: o4-mini improves slightly with summaries (51.0% -> 53.2%), GPT 4.1 shows smaller difference (~42-44%), while Gemini 2.5 Flash performance is hurt by summaries (45.0% -> 37.3%). We note that LLM-generated code summaries can themselves be detail-heavy and potentially more confusing than the original code, especially when attempting to describe low-level implementation logic in natural language, which may explain why summaries do not consistently improve performance and can even hurt it. We will add these ablation results to Appendix B and discuss the implications for future work on optimizing model performance on SpecGen tasks.
>
> | Model | No Ref | GT Code Ref | Gen Code Ref | GT Code Summary Ref | Gen Code Summary Ref |
> |-------|--------:|-------------:|--------------:|------------------:|-----------------:|
> | o4-mini | 51.0% | 50.6% | 50.7% | 53.2% | 52.6% |
> | GPT 4.1 | 42.6% | 43.9% | 41.6% | 43.9% | 41.8% |
> | Gemini 2.5 Flash | 45.0% | 43.3% | 45.8% | 37.3% | 37.4% |
>
> Where "No Ref" means specification generation without code reference, "GT Code Ref" uses ground truth code as reference, "Gen Code Ref" uses LLM-generated code as reference, "GT Code Summary Ref" uses ground truth code summary as reference, and "Gen Code Summary Ref" uses generated code summary as reference.

---

> ### Author Response · Authors · 2025-11-21
>
> **W1: Does VERINA’s current size and scope already represent a valuable contribution?**
>
> We agree with the reviewer that extending VERINA in terms of size and scope would be valuable, though such extensions require substantial efforts. We therefore consider them as important future work, and will expand our discussion accordingly. That said, we believe VERINA already presents significant values in its current form.
>
> Regarding benchmark size, VERINA includes 189 manually curated instances, comparable to widely used benchmarks such as HumanEval [1] (164 problems), which have successfully measured progress in code generation. VERINA is similarly designed to benchmark advances in verifiable code generation. As discussed in Q4, the performance differences observed between VERINA-A and VERINA-B demonstrate that VERINA captures meaningful difficulty strata and can effectively differentiate model capabilities. Moreover, we intentionally prioritized benchmark quality over quantity: each instance requires establishing ground truth code implementations and specifications, as well as comprehensive test suites. This level of rigor ensures VERINA can serve as a robust and trusted benchmark for the community.
>
> Regarding scope, although VERINA’s tasks may appear modest, our evaluation demonstrates that today’s LLMs still struggle significantly, especially on ProofGen, highlighting core limitations that current techniques must address before scaling to larger real-world programs. As such, VERINA establishes an essential milestone toward scalable and practical verifiable code generation, making it a timely and valuable contribution to the field.
>
> [1] Chen et al., Evaluating Large Language Models Trained on Code. arXiv:2107.03374.

---

> ### Author Response · Authors · 2025-11-28
>
> Thank you for the constructive feedback. We have provided detailed responses to all your questions and have conducted the additional experiments you requested. We would greatly appreciate any additional feedback you might have on our responses.

---

> ### Author Response · Authors · 2025-12-03
>
> We have revised our manuscript to address all the mentioned concerns. The changes are highlighted in blue. Specifically, we did the following changes (all line numbers are from the revised version):
> - Q1&W6: We added the experiment results and discussion in Appendix C.5 line 1302-1310 and Table 6;
> - Q2&W2: We added clarifying sentences in Section 3.2 line 237-242;
> - Q3&W3-5: We added Appendix B to discuss data contamination risk and mitigation;
> - Q4: We added the experiment results and discussion in Appendix D line 1508-1533 and Figure 17;
> - Q5: We added clarifying sentences in Appendix C.3 line 927-930;
> - Q6: We added the experiment results and discussion in Appendix D line 1455-1462 and Table 7;
> - W1: We added clarifying sentences in Section 6 527-533 and also Appendix A to discuss VERINA’s scope.

---

### Official Review · Reviewer_sru5 · 2025-11-01

**Soundness:** 3
**Presentation:** 2
**Contribution:** 3
**Rating:** 6
**Confidence:** 4

**Summary:**

The paper introduces VERINA, a benchmark for verifiable code generation built on the Lean 4 proof assistant. VERINA provides a framework for evaluating three core tasks: code generation (CodeGen), formal specification generation (SpecGen), and proof generation (ProofGen), as well as their modular compositions. The benchmark consists of 189 manually curated problems, each with a natural-language description, reference implementation, formal specification, and a test suite. The authors propose a hybrid evaluation pipeline for SpecGen that combines automated proof attempts with property-based testing. An extensive evaluation of general-purpose LLMs and specialized theorem provers reveals that ProofGen is the primary bottleneck in the end-to-end pipeline, with even iterative refinement showing limited success.

**Strengths:**

1. The paper makes a well-scoped and valuable contribution by introducing a unified benchmark for code, specification, and proof generation.
2. The proposed hybrid evaluation metric for SpecGen is a practical approach to a difficult problem, leveraging both formal proving and extensive testing to assess specification quality.
3. The paper provides a thorough experimental evaluation across a diverse set of models, including general-purpose LLMs, specialized provers, and an agentic baseline, offering a snapshot of current capabilities.

**Weaknesses:**

1. Gaps in Evaluation Metrics: The SpecGen evaluation logic is potentially not perfect. It uses negative test cases to evaluate both the soundness and completeness of different specification components, but does not appear to distinguish between tests that violate the pre-condition versus those that violate the post-condition. This can lead to noisy or incorrect soundness/completeness judgments.
2. The paper's claims of test suite adequacy rely on coverage metrics from Python reference implementations, as Lean lacks mature coverage tooling. This is a significant gap, as coverage in one language does not guarantee equivalent coverage in a translated Lean implementation. Can authors discuss more about it?
3. The evaluation lacks a publicly available frontier reasoning model (e.g., OpenAI's o3 or GPT-5, Google's gemini-2.5-pro or Claude-4/4.1 opus), while it currently claims o4-mini to be a frontier model. While the conclusions may not be wildly different, given this is a benchmark paper, evaluating frontier models remains essential.
4. While LLM-based provers are part of the SpecGen evaluator, their extremely low success rates (<4%) and inherent non-determinism make their inclusion in the primary metric questionable. Some manual evaluations may either strengthen the claim or clarify the limitations.


**Other Minor Weaknesses**
1. Despite the emphasis on compositionality, the paper omits results for the full end-to-end task (Description → Code + Spec + Proof). Reporting this, along with the success rate for generated code satisfying generated specifications, wuld further strengthen the paper.
2. Insufficient Failure Analysis: The qualitative case studies of failure modes are useful, but the paper would be significantly strengthened by a quantitative breakdown of error categories (e.g., syntax errors, API hallucinations, tactic misuse, logical errors). Correlating these failure types with the dataset origin (VERINA-A vs. VERINA-B) would provide more actionable insights for future research.
3. Limited Scope and Generalizability: The benchmark's reliance on Lean and the Interactive Theorem Proving (ITP) paradigm necessarily limits the external validity of its findings to other verification ecosystems. Furthermore, the paper could characterize the classes of real-world programs (e.g., those with effects, IO, or concurrency) that are out of scope for the current benchmark design.
4. Presentation and Framing Improvements: a.) > "The high-quality samples and robust metrics"  in the introduction. Robust metrics have not been defined yet, or clarified why exactly they are robust. b.) "top-performing general-purpose LLM" is referred to o4-mini. c.) "ITPs support constructing proofs with explicit intermediate steps. This visibility enables LLMs to diagnose errors, learn from unsuccessful steps, and iteratively refine their proofs," in related work. However, iterative refinement is also possible with ATP.

**Questions:**

Please see the Weaknesses above. In addition:

1. The paper states the SpecGen evaluator compares outcomes for R and ~R to select the "more accurate" result. Can you please clarify what "more accurate" means here?

---

> ### Author Response · Authors · 2025-11-21
>
> We thank the reviewer for their insightful and constructive review and address their questions and feedback below. We will post a revised manuscript on OpenReview based on the feedback and additional experiments within the next few days. (`Wi` stands for weakness `i`; `MWi` stands for minor weakness `i`; `Qi` stands for question `i`)
>
> **W1: Does SpecGen evaluation properly distinguish between negative tests that violate pre-conditions versus post-conditions?**
>
> Yes. To clarify, we do explicitly distinguish between negative tests that violate pre-conditions and those that satisfy pre-conditions but violate post-conditions. Figure 1 has showcased such negative tests: Line 24 is a negative test that violates the pre-condition and Line 25-27 are negative tests that only violate the post-condition. Our evaluation applies these tests separately to assess generated pre-conditions and post-conditions. We have illustrated how we applied these tests in Figures 10 and 11 of Appendix A.5. We will revise Sections 3.2 and 4.1 to clearly explain that we distinguish negative tests for pre- and post-conditions, and we apply them separately during evaluation.
>
> **W2: Does your test suite achieve 100% coverage in Lean reference implementations?**
>
> Yes. We acknowledge the reviewer’s concern that achieving 100% coverage in Python does not theoretically guarantee the same in Lean. To validate this in practice, we manually inspected all benchmark instances and executed our test suites on the Lean ground-truth implementations. We confirmed that our test suites also achieve 100% coverage in our Lean ground truth code. This is feasible because, for the problems we consider, cross-language test suites are typically sufficient to assess functional correctness across language implementations, which is a common practice adopted by LeetCode and prior work on evaluating LLM-generated code [1, 2]. Our additional manual inspection further strengthens the benchmark quality. We will revise the paper to clarify this.
>
> [1] Cassano et al. MultiPL-E: A Scalable and Extensible Approach to Benchmarking Neural Code Generation. IEEE TSE, 2023.
>
> [2] Roziere et al. Leveraging Automated Unit Tests for Unsupervised Code Translation. ICLR, 2022.
>
> **W3: Please evaluate frontier reasoning models.**
>
> We have conducted additional experiments with OpenAI o3 and Claude 4 Opus. The results are shown in the following table.
> | Model | CodeGen | SpecGen (Sound & Complete) | ProofGen |
> |:-------|:---------|:---------------------------|:----------|
> | OpenAI o3 | 72.6% | 52.3% | 4.9% |
> | Claude 4 Opus | 67.3% | 50.0% | 0.0% |
> | OpenAI o4-mini (baseline) | 61.4% | 51.0% | 3.6% |
>
> From the results, we can conclude verifiable code generation remains highly challenging, even for frontier models. While CodeGen performance improves noticeably, SpecGen shows limited gains, and ProofGen remains the primary bottleneck. We will update the manuscript with these results and revise references to "best performing models" accordingly.
>
> **W4: Are LLM-based provers appropriate for the SpecGen evaluation, given their current low success rates and non-determinism?**
>
> We believe yes. Although current LLMs have relatively low success rates in proving specification equivalence, we include them because any successful proofs offer formal guarantees. Moreover, LLMs’ proof capabilities and stability are rapidly improving with newer prover models, stronger proof search agents, and new automation tactics like grind. This will make our SpecGen metric increasingly powerful over time. When the proving step fails, our testing-based evaluator always serves as a fallback. Overall, this comprehensive design ensures that SpecGen remains robust today while progressively delivering stronger formal guarantees as model capabilities improve. We will add a clarification to this.

---

> > ### Author Response · Authors · 2025-11-21
> >
> > **MW1.1: Can you report results for the full end-to-end task?**
> >
> > Yes. Below, we show the end-to-end verifiable code generation results where models generate code, specification, and proof. The end-to-end results show similar patterns to individual task performance, where ProofGen tasks remain the primary bottleneck in verifiable code generation. We will add full end-to-end results to the manuscript.
> >
> > | Model | Code Pass Rate | Spec Pass Rate | Code+Spec Pass Rate | E2E Pass Rate |
> > |-------|---------------:|---------------:|--------------------:|--------------------------:|
> > | o4-mini | 60.43% | 45.45% | 32.62% | 3.21% |
> > | GPT 4.1 | 56.68% | 39.57% | 29.41% | 2.67% |
> > | Claude Sonnet 3.7 | 42.25% | 41.18% | 22.99% | 1.60% |
> > | Gemini 2.5 Flash | 37.97% | 40.64% | 26.74% | 2.14% |
> >
> > **MW1.2 What is the success rate for generated code satisfying generated specifications?**
> >
> > Out of 189 tasks, we observed that o4-mini successfully generates proofs for generated code and specifications in 0.86% of cases on average where either the code or specification is incorrect. Similarly, Gemini 2.5 Flash has 0.54% of cases where proofs verify but the code or specification is wrong. We did not observe this phenomenon in other models. We will revise the manuscript to include these results.
> >
> > **MW2: Can you provide a quantitative breakdown of error categories with correlation to the dataset origin?**
> >
> > We have conducted a quantitative breakdown of failure categories across models and dataset origins (VERINA-A vs VERINA-B). Here is the failure breakdown of o4-mini on ProofGen task:
> > | Category | VERINA-A | VERINA-B | Overall |
> > |----------|----------:|----------:|----------:|
> > | Cheat Code Usage | 18.55% | 40.25% | 28.30% |
> > | Incomplete Proof (unsolved goals) | 13.91% | 12.10% | 13.10% |
> > | Unknown Import | 2.82% | 0.74% | 1.89% |
> > | Syntax Error | 3.23% | 4.20% | 3.66% |
> > | Tactic Failed | 8.27% | 4.44% | 6.55% |
> > | Type Mismatch | 1.61% | 1.73% | 1.66% |
> > | Unknown Constant | 7.46% | 3.21% | 5.55% |
> > | Unknown Identifier | 12.30% | 12.35% | 12.32% |
> > | Unknown Tactic | 20.77% | 14.07% | 17.76% |
> > | Other | 11.09% | 6.91% | 9.21% |
> >
> > This reveals that on harder problems (VERINA-B), models resort more frequently to "cheat codes" (like sorry) rather than attempting complete proofs, while on simpler problems they make more tactical and identifier errors in genuine proof attempts.
> >
> > We have already conducted quantitative breakdown of CodeGen and SpecGen error categories in Tables 6, 7, and 8 in Appendix B. We will add the dataset-origin-correlated breakdown (VERINA-A vs VERINA-B) for these error categories in the revised manuscript, as well as the full proof error category analysis.
> >
> > **MW3: Can you comment on the generalizability and scope of VERINA?**
> >
> > We agree with the reviewer that extending VERINA to additional verification ecosystems and more complex real-world programs would be valuable, though such extensions require substantial efforts. We therefore consider these important directions for future work, and will expand our discussion accordingly. That said, we believe VERINA already presents significant values in its current form.
> >
> > Regarding VERINA’s generalizability, while Lean’s syntax differs from other ecosystems such as Dafny and Verus, the core challenges in verifiable code generation are shared. Both ITP and ATP frameworks require: (i) generating correct code and sound/complete specifications, and (ii) identifying key properties such as loop invariants for constructing proofs. The insights from VERINA on these core capabilities are therefore broadly relevant.
> >
> > Moreover, even with its current scope, VERINA uncovers important limitations of state-of-the-art LLMs, e.g., on ProofGen. This means current techniques must succeed on VERINA before scaling to larger, real-world programs. As such, VERINA establishes an essential milestone toward scalable and practical verifiable code generation, making it a timely and valuable contribution to the field.
> >
> > **MW4: Presentation improvements.**
> >
> > We thank the reviewer for these helpful suggestions and will address them in the revised manuscript.
> >
> > **Q1: Can you clarify what "more accurate" means?**
> >
> > “More accurate’’ simply means that in some cases the evaluator returns unknown for `R`, but `¬R` can be decided as `True` or `False` (or vice versa). In those situations, we take the definitive outcome as the final result. We will clarify this in the revised manuscript.

---

> ### Author Response · Authors · 2025-11-28
>
> Thank you for the constructive feedback. We have provided detailed responses to all your questions and have conducted the additional experiments you requested. We would greatly appreciate any additional feedback you might have on our responses.

---

> ### Author Response · Authors · 2025-12-03
>
> We have revised our manuscript to address all the mentioned concerns. The changes are highlighted in blue. Specifically, we did the following changes (all line numbers are from the revised version):
> - W1: We added clarifying sentences in Section 3.2 line 250-252 and Section 4.1 line 315-318;
> - W2: We added clarifying sentences in Section 3.2 line 237-242;
> - W3: We added the evaluation result of OpenAI o3 and Claude Opus 4.0 to relevant figures in Section 5. We also updated the abstract and introduction;
> - W4: We added clarifying sentences in Section 4.1 line 309-310 and Appendix C.5 line 1299-1301;
> - MW1: We added the end-to-end result and discussion in Section 5 line 435-452;
> - MW2: We added the quantitative breakdown of code/spec/proof failure mode categorized by VERINA-A and VERINA-B to Appendix D. We updated/added line 1562-1570 and Table 8-19.
> - MW3: We added Appendix A to discuss the scope and generalizability of VERINA;
> - MW4: We have revised the mentioned presentation;
> - Q1: We added clarifying sentences in Section 4.1 line 348-350.

---

### Author Response · Authors · 2025-12-03

Dear Chairs,

We acknowledge the decision to revert review scores and close the author-reviewer discussion due to the OpenReview security incident. We sincerely appreciate your efforts in managing ICLR during this challenging situation. To facilitate the upcoming review process, we would like to summarize how our rebuttal addressed all review concerns and strengthened our paper.

Reviewer sru5: We have addressed all concerns including additional frontier models evaluation, evaluation metrics clarification, test suite coverage adequacy validation, systematic proof failure categorization, and presentation improvements.

Reviewer 6DqP explicitly stated "**Addressing these issues may lift the work to acceptance**", and we have comprehensively addressed all their concerns. This includes mitigating the data contamination risk by checking N-gram overlap with pre-training dataset, test suite coverage adequacy validation, spec metric sensitivity analysis.

Reviewer a4Ez: Addressed all concerns including Lean justification with production use cases (AWS Cedar, LNSym, Aeneas, Clear), systematic proof failure categorization, complexity statistics, and corrected citations.

In summary, VERINA provides a rigorously validated benchmark for verifiable code generation with novel evaluation metrics, and our experiments reinforce the main claims while offering actionable insights for future research.

Best,

Authors

---

### Meta-Review · Area_Chair_e7cq · 2026-01-07

**Summary:**

The paper introduces VERINA, a benchmark for evaluating verifiable code generation, focusing on code generation, specification generation, and proof generation. It features 189 manually curated tasks in Lean and evaluates state-of-the-art models. Reviewers appreciate its comprehensive approach but express concerns about the benchmark's limited scope, potential data contamination, and the generalizability of the Lean-based evaluation. Some reviewers also point to the lack of systematic analysis of proof generation failures. Despite these weaknesses, the authors have addressed many concerns in their rebuttal and revisions. Given the significant contribution and improvements made, I recommend accepting the paper, though future work on expanding scope and mitigating data contamination would strengthen its impact.

**Reviewer Concerns:**

Concerns Addressed by the Rebuttal:

1. Evaluation completeness and rigor: The authors added evaluations on frontier models (e.g., OpenAI o3, Claude Opus) and full end-to-end results, directly addressing concerns about missing key experiments.

2. SpecGen and ProofGen analysis: The rebuttal provided detailed quantitative breakdowns of failure modes and sensitivity analyses, substantially strengthening the empirical support.

3. Data contamination risk: N-gram overlap analysis and clarification of ProofGen isolation convincingly mitigated major contamination concerns.

Concerns Still Outstanding:

1. Benchmark scope and scale: Despite justification, the dataset size (189 tasks) and focus on single-file Lean programs still limit representativeness for large-scale, real-world systems.

2. Generalizability beyond Lean: While the authors argue conceptual relevance to other ecosystems, the lack of ATP-based benchmarks (e.g., Dafny, Verus) remains an open limitation.

**Reviewer Scores:**

Reviewer sru5: Likely to increase. Most technical concerns (frontier models, SpecGen evaluation clarity, end-to-end results, failure analysis) were directly addressed.

Reviewer 6DqP: Likely to increase. The rebuttal substantially mitigated the key concerns on data contamination, coverage validity, and interpretability, though some reservations about benchmark scope may remain.

Reviewer a4Ez: Likely to increase. The authors addressed Lean justification, proof failure analysis, task statistics, and corrected citations, but concerns about generalizability and task simplicity may still limit enthusiasm.

---

### Decision · Program_Chairs · 2026-01-26

Accept (Poster)